# A temperature-regulated circuit for feeding behavior

Shaowen Qian [1,2✉], Sumei Yan[1], Ruiqi Pang[1,3], Jing Zhang[4], Kai Liu[2], Zhiyue Shi[1], Zhaoqun Wang[1], Penghui Chen[1], Yanjie Zhang[1], Tiantian Luo[1], Xianli Hu[1], Ying Xiong [1✉] & Yi Zhou [1✉]

Both rodents and primates have evolved to orchestrate food intake to maintain thermal homeostasis in coping with ambient temperature challenges. However, the mechanisms underlying temperature-coordinated feeding behavior are rarely reported. Here we find that a non-canonical feeding center, the anteroventral and periventricular portions of medial pre-optic area (apMPOA) respond to altered dietary states in mice. Two neighboring but distinct neuronal populations in apMPOA mediate feeding behavior by receiving anatomical inputs from external and dorsal subnuclei of lateral parabrachial nucleus. While both populations are glutamatergic, the arcuate nucleus-projecting neurons in apMPOA can sense low temperature and promote food intake. The other type, the paraventricular hypothalamic nucleus (PVH)-projecting neurons in apMPOA are primarily sensitive to high temperature and suppress food intake. Caspase ablation or chemogenetic inhibition of the apMPOA→PVH pathway can eliminate the temperature dependence of feeding. Further projection-specific RNA sequencing and fluorescence in situ hybridization identify that the two neuronal populations are molecularly marked by galanin receptor and apelin receptor. These findings reveal unrecognized cell populations and circuits of apMPOA that orchestrates feeding behavior against thermal challenges.

[1] Department of Neurobiology, Chongqing Key Laboratory of Neurobiology, School of Basic Medicine, Army Medical University, Chongqing, China. [2] Department of Medical Imaging, The 960th Hospital of Joint Logistics Support Force of PLA (Former Jinan Military General Hospital), Jinan, Shandong, China. [3] Advanced Institute for Brain and Intelligence, School of Medicine, Guangxi University, Nanning, Guangxi, China. [4] Department of Radiation Oncology, Tianjin Medical University Cancer Institute and Hospital, National Clinical Research Center for Cancer, Tianjin, China. ✉email: qianshaowen1110@163.com; xiongying2001@163.com; zhouyisjtu@gmail.com

F eeding behavior is regulated by multiple hard-wired neural circuits that underlie diverse internal homeostatic factors and external environmental factors[1–5]. Ambient temperature, as a pivotal homeostatic-related environmental factor, intimately orchestrate feeding behavior[5–8]. To maintain homeostasis for survival, both rodents and primates have evolved to optimally orchestrate food intake in sensing ambient thermal challenges[9–13]. A long-standing theory for feeding behavior proposes a "thermostatic" mechanism, in which food intake acts as a mechanism of thermoregulation to maintain proper energy intake[9]. Under low ambient temperature, the higher demand for heat production forces body to further increase energy intake, which is manifested as a promoted feeding. In the instances of extremely hot environments, the lower thermal demand and higher dissipation load preferentially suppress food intake, overriding the dietary state[9,14,15]. However, the neural mechanisms underlying the ambient temperature-coordinated feeding behavior are rarely reported.

The neuronal populations in medial preoptic area (MPOA) have been identified to be responsive to ambient temperature, including subpopulations of neurons that express leptin receptor[8,16], adenylate cyclase-activating peptide 1, brain-derived neurotrophic factor[17,18], and pyroglutamylated RFamide peptide[19]. Intermingled with thermosensory aspects, the activations of these neurons also resulted in changes in feeding behavior[8,16,19]. Local thermal manipulations (e.g., heating, cooling, or lesions) on the MPOA in goats, rats or pigs would cause rapid feeding behavior or cessation, regardless of their original dietary states[14,15,20–24]. Genetically activating MPOA neurons even could cause mice to enter a state of hibernation or torpor, showing hypothermia and anorexia[18,19,25]. Pathologically, patients with hypothalamo-pituitary deficiency (e.g., craniopharyngioma, pituitary tumor)[26–28], as well as animals with lesions in the preoptic area and anterior hypothalamus (POA/AH)[15,21,29], usually exhibit abnormalities in autonomic thermoregulation along with abnormal feeding behavior. Recently, anterograde labeling of MPOA neurons revealed dense projections to multiple appetite centers, such as the bed nucleus of the stria terminalis (BNST), arcuate nucleus (ARC), paraventricular hypothalamic nucleus (PVH), lateral hypothalamus (LH)[30,31]. It would be interesting to investigate whether thermosensitive MPOA neurons and their specific circuits exert neural regulations on appetite centers in coping with ambient thermal challenges.

To address these issues, we dissected the contribution of POA/AH to temperature-dependent feeding behavior. We found that among multiple thermosensitive subregions of the POA/AH, two populations of glutamatergic neurons in the anteroventral and periventricular portions of medial preoptic area (apMPOA) were responsive to altered dietary states, and orchestrated two distinct pathways to mediate feeding behavior in receiving anatomical inputs from external and dorsal subnuclei of lateral parabrachial nucleus (LPB). The ARC-projecting neurons in apMPOA could sense low temperature, and promote food intake. Whereas, the PVH-projecting neurons in apMPOA were primarily sensitive to high temperature and suppressed food intake. Further projection-specific RNA sequencing identified that the two neuronal populations were molecularly marked by galanin receptor and apelin receptor. Collectively, our results highlight previously unrecognized cell populations and circuits of apMPOA that orchestrates feeding behavior in coping with ambient temperatures.

## Results

### Glutamatergic neurons in the apMPOA respond to altered dietary states.
We first identified the behavioral pattern of temperature-dependent feeding behavior (Fig. 1a, b and Supplementary Fig. 1). We observed that 2 h of hot exposure reduced the food intake of the mice by 25.1%, whereas 2 h of cold exposure increased food intake by 20.3% (Fig. 1b). The temperature-dependence of food intake persisted under longer thermal exposure of 12 h (Supplementary Fig. 1a). Ambient exposures did not result in rectal temperature changes (Supplementary Fig. 1b).

Next, we sought to determine whether central thermosensitive nuclei were involved in the phenotype of temperature-regulated feeding behavior. We deprived the mice of food for 12 h and quantified C-Fos expression in the hypothalamic nuclei. We observed substantial C-Fos expressions in the anteroventral periventricular portions of the medial preoptic area (apMPOA), including ventromedial POA (VMPO), anteroventral periventricular nucleus (AVPe), ventral median preoptic nucleus (MnPO), preoptic part of periventricular hypothalamic nucleus (PVpo) and anterior MPOA among multiple nuclei within the POA/AH, but not the lateral preoptic area (LPO), AH, medial preoptic nucleus (MPN), or lateral parabrachial nucleus (LPB) (Fig. 1c and Supplementary Fig. 2). Mice that were fed ad libitum or underwent refeeding for 2 h after fasting showed no C-Fos expression in the apMPOA (Fig. 1d, e and Supplementary Fig. 3a, b). The apMPOA is enriched with massive thermosensitive neurons with substantial C-Fos activated at low and high ambient temperatures but not at neutral temperature (Supplementary Fig. 3a, d, g). At neutral temperature, fasting resulted in substantial C-Fos expression in apMPOA but disappeared after refeeding, suggesting that these C-Fos+ neurons in apMPOA respond to altered dietary status. (Fig. 1d, e and Supplementary Fig. 3a–c). To identify the cell types with fasting-induced C-Fos, we bred vGluT2-cre and GAD2-cre mice with Ai9 reporter mice, in which the vGluT2+ and GAD2+ neurons were labeled with tdTomato (Fig. 1g, h). The fasting-induced C-Fos in the apMPOA exhibited substantial colocalization with vGluT2+ neurons. Approximately 60% of C-Fos+ neurons expressed vGluT2, whereas nearly 65% of vGluT2+ neurons expressed C-Fos (Fig. 1g, i, j). In the GAD2-cre::Ai9 mice, fasting-induced C-Fos exhibited minor overlap with the GAD2 (20% C-Fos+ neurons expressed GAD2) (Fig. 1h–j).

To further clarify the neuronal responses of apMPOA to dietary state, we injected activity-dependent GCaMP6s into the apMPOA of vGluT2-cre and GAD2-cre mice and installed an optical fiber above the apMPOA to record calcium signal during feeding (Fig. 1k). We found no significant calcium responses to the feeding in GAD2-cre mice during 15 min recordings (Fig. 1l–n). In contrast, vGluT2-cre mice showed a substantial calcium signal decrease in 5 min after the onset of feeding, indicating that vGluT2+ neuron activity decreased substantially after the onset of feeding (Fig. 1m, n). Moreover, the decreased calcium response of glutamatergic apMPOA neurons did not occur upon sensory detection of food, or non-edible object (such as tearing paper), and did not recover after pausing feeding behavior (Supplementary Movie 1). These findings indicated that glutamatergic apMPOA neurons, but not GABAergic neurons, responded to altered dietary states, such as hunger-activated response if overnight fasted, and satiety-inhibited response if refed.

### Optogenetic activation on glutamatergic apMPOA neurons modulates body temperature.
We next targeted the expression of channelrhodopsin-2 (ChR2) to glutamatergic and GABAergic apMPOA neurons to explore their regulatory roles in feeding (Fig. 2a, b). Remarkably, the food intake of apMPOA$^{vGluT2-ChR2}$ stimulated mice was significantly decreased by 89.1% compared with that of the mice receiving no opto-stimulation (Fig. 2c), while the light stimulation on apMPOA$^{GAD2-ChR2}$ neurons

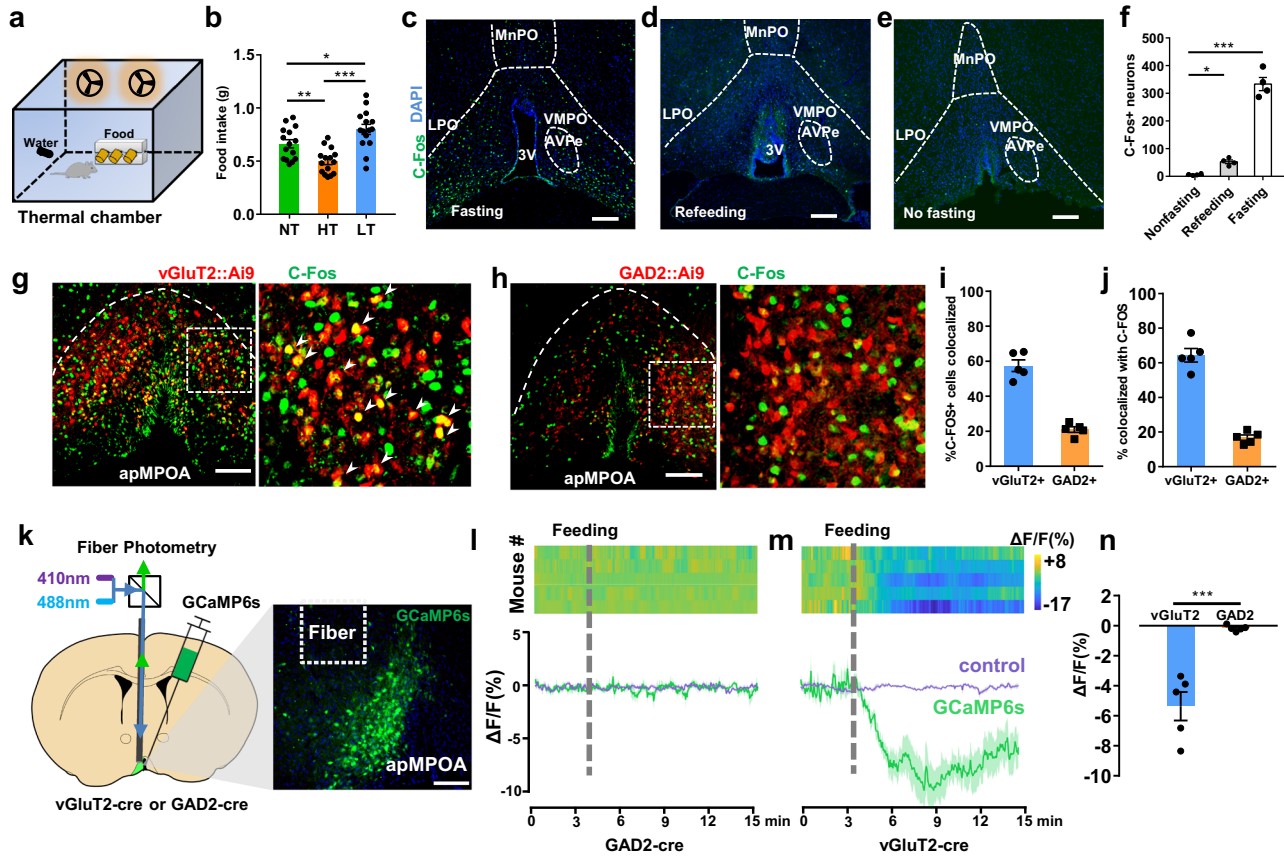

**Fig. 1 Glutamatergic apMPOA neurons respond to altered dietary states. a** Scheme of feeding test in a temperature-controlled chamber. **b** Differences of 2 h food intake among the three ambient temperatures. NT neutral temperature, HT high temperature, LT low temperature. One-way ANOVA and post hoc least significant difference (LSD) multiple comparison, $*p = 0.019$, $**p = 0.004$, $***p < 0.001$, $n = 15$ animals for each group. **c–f** C-Fos expressions in the apMPOA in three dietary states: 12 h fasting, 2 h refeeding and no fasting. AVPe, anteroventral periventricular nucleus; VMPO, ventromedial preoptic nucleus; MnPO, median preoptic nucleus; 3V, the 3rd ventricle. Scale bar, 100 μm. One-way ANOVA and post hoc LSD multiple comparison, $*p = 0.048$, $***p < 0.001$, $n = 4$ animals for each group. **g**, **h** Colocalizations of fasting-induced C-Fos (green) with vGluT2+ (red) and GAD2+ neurons (red) in vGluT2::Ai9 (**g**) and GAD2::Ai9 mice (**h**). Images were obtained from transgenic mice by crossing Ai9 (Cre-dependent tdTomato reporter) with vGluT2-cre (**g**), and Ai9 with GAD2-cre (**h**). Scale bar, 100 μm. **i** Percentage of C-Fos+ neurons that are vGluT2+ and GAD2+. $n = 5$ animals for each group. **j** Percentage of vGluT2+ and GAD2 + neurons that express C-Fos. $n = 5$ animals for each group. **k** Scheme of fiber photometry setup, and representative coronal section of apMPOA neurons expressing GCaMP6s. Image representative of $n = 5$ mice. Scale bar, 100 μm. **l**, **m** GcaMP6s (mean, green line; SEM, green shading) and control (mean, purple line; SEM, purple shading) signal changes ($\Delta F/F$) in response to feeding in GAD2-cre (**l**) and vGluT2-cre mice (**m**) after 24 h fasting. $n = 5$ animals for each group. **n** The mean $\Delta F/F$ in a 5 min window after the onset of feeding. T test (two-sided), $***p < 0.001$, $n = 5$ animals for each group. All error bars show SEM.

caused no significant change in food intake (Fig. 2c). Consistent with previous studies[8,32], as a thermosensitive and regulatory center, the opto-stimulation on apMPOA[vGluT2-ChR2] neurons, but not apMPOA[GAD2-ChR2] neurons, induced apparent autonomic thermoregulation through inhibited brown adipose tissue (BAT) thermogenesis, as well as enhanced heat dissipation via tail vasodilation (Fig. 2d, e). Notably, we found that the activation of apMPOA[vGluT2-ChR2], but not apMPOA[GAD2-ChR2], led to a remarkable cold defensive behavior, nest building (Supplementary Movies 2 and 3), which might indirectly affect food intake. The apMPOA comprises multiple cell types and downstream projections that mediate complex innate behaviors[30,33], and homeostatic regulatory processes[17,34]. Therefore, here we could not conclude that direct activation on the apMPOA[vGluT2-ChR2] suppressed food intake, which awaited further dissecting downstream regions of apMPOA[vGluT2-ChR2] neurons.

**Optogenetic activations on the glutamatergic apMPOA-to-ARC/PVH pathways orchestrate food intake.** We next dissected the downstream targets of apMPOA[vGluT2+] neurons to explore the potential regulatory effect on food intake. We performed anterograde tracing by injecting AAV-DIO-hChR2-EYFP into the apMPOA of vGluT2-cre mice (Fig. 3a). Consistent with previous studies[30,31], and data from the Allen Brain Atlas (Experiment 113554719, http://connectivity.brain-map.org/), we found that apMPOA[vGluT2+] neurons exhibited widespread projections to multiple brain regions involved in feeding behavior, including the BNST, LH, tuberal nucleus (TN), PVH, dorsomedial hypothalamic nucleus (DMH), ARC, paraventricular thalamic nucleus (PVT), basolateral and basomedial amygdala (BA), ventral tegmental area (VTA) and periaqueductal gray (PAG) (Fig. 3b and Supplementary Fig. 4b). We quantified neurons directly under apMPOA[vGluT2+] axons that expressed hunger-induced C-Fos. The number of neurons expressing C-Fos was increased in several downstream regions, including PVH, ARC, LH, DMH, BNST, PAG (Fig. 3b), but not in the others, including PVT, TN, BA, and VTA (Supplementary Fig. 4b).

Among the downstream regions of apMPOA[vGluT2+] neurons, several were previously reported to be involved in regulating

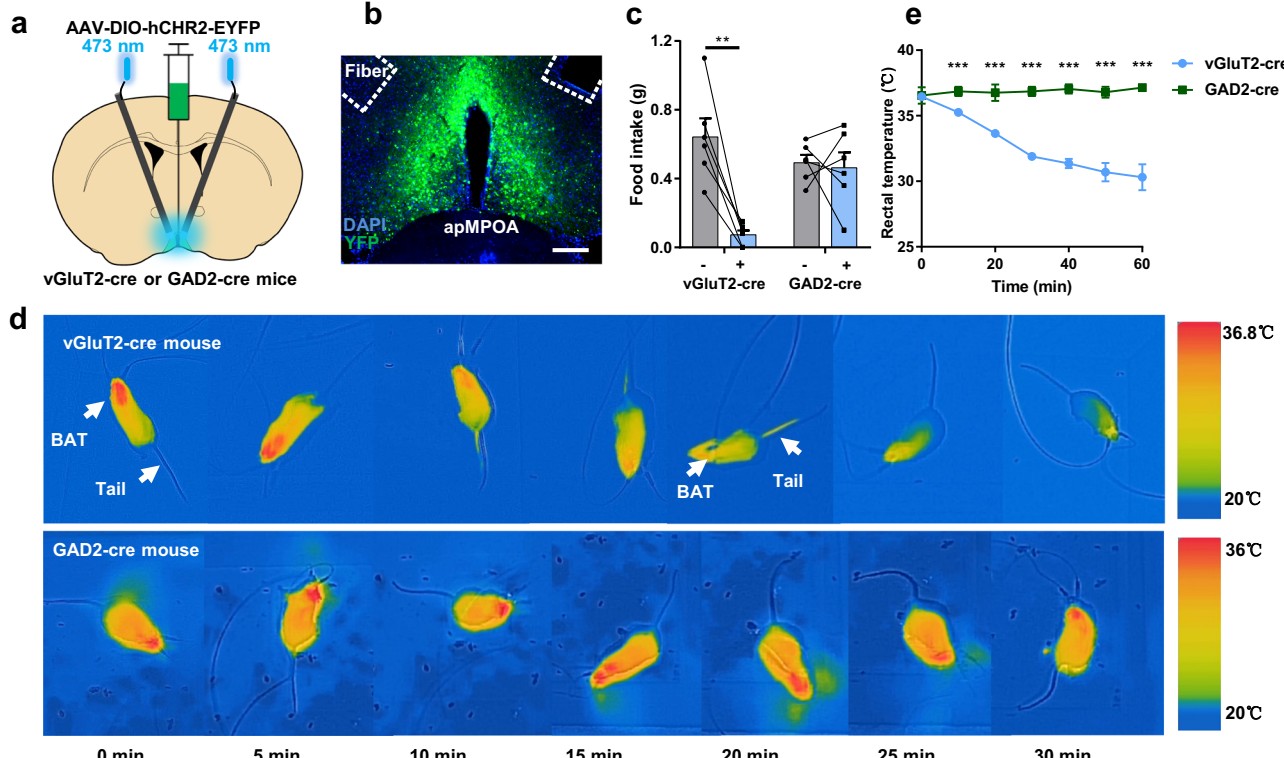

**Fig. 2 Opto-stimulation on glutamatergic apMPOA neurons induces decreases in food intake and rectal temperature. a, b** Schematic and representative coronal images of injecting AAV virus expressing ChR2 into the apMPOA in vGluT2-cre and GAD2-cre mice and oblique optical fiber implantation above the apMPOA. Image representative of $n = 10$ mice. Scale bar, 100 μm. **c** Opto-stimulation (2 h) on the apMPOA neurons in vGluT2-cre mice decreased food intake, but not in GAD2-cre mice. Paired t tests (two-sided), $**p = 0.003$, $n = 6$ animals for each group (a total of 10 mice models were made, and 4 mice for each group were excluded due to missed injections or death). **d** Infrared thermography revealed that opto-stimulation (30 min) on the apMPOA neurons in vGluT2-cre mice, but not GAD2-cre mice, promoted tail heat dissipation and inhibited BAT thermogenesis. **e** Opto-stimulation (60 min) on the apMPOA neurons in vGluT2-cre mice induced transient rectal temperature decrease (record 60 min for 7 time points), but not in GAD2-cre mice (Rectal temperature of vGluT2-cre mice was significantly lower than that of GAD2-cre mice at the points 2-7, all $***p < 0.001$). ANOVA interaction effect: $F_{(6, 5)} = 34.45, ***p < 0.001$; main effect of time: $F_{(6, 5)} = 30.45, ***p < 0.001$; main effect of group: $F_{(1, 10)} = 641.33, ***p < 0.001$. Within-group tests showed significant simple effect of time in vGluT2-cre group ($F_{(6, 5)} = 64.14, ***p < 0.001$), but not in the GAD2-cre group ($F_{(6, 5)} = 0.77, p = 0.63$). Two-way repeated-measure ANOVA, $n = 6$ animals for each group. All error bars show SEM.

feeding behavior[35–38], raising the hypothesis that these apMPOA-projecting neural pathways are probably involved in temperature-regulating feeding behavior. We performed a systematic manipulation on each downstream region of apMPOA$^{vGluT2+}$ neurons (Fig. 3c and Supplementary Fig. 4a). We discovered that optogenetic activation on apMPOA→ARC terminal fibers increased food intake by 56.8% (Fig. 3d). In contrast, terminal activation on the PVH reduced food intake by 78.1% (Fig. 3d). The regulatory effects on food intake of terminal activations on both ARC and PVH were robust under high and low ambient temperature (Fig. 3e, f), and did not cause changes in rectal temperature (Supplementary Fig. 4i) or cold defensive behavior mentioned above. These findings indicated that glutamatergic apMPOA neurons could specifically orchestrate feeding behavior through downstream regions ARC and PVH. We also expressed ChR2 in the apMPOA-recipient ARC and PVH neurons by injecting AAV1-cre into the apMPOA and AAV-DIO-ChR2 into the ARC and PVH. Optogenetic activation of apMPOA-recipient ARC and PVH neurons resulted in similar effects to stimulating apMPOA→ARC/PVH terminals: increased food intake in the ARC-stimulating mice, but reduced food intake in the PVH-stimulating mice (Supplementary Fig. 5). Additionally, terminal activation on the DMH reduced food intake by 84.5% (Fig. 3d). However, the mice exhibited apparent nest building, a cold defensive behavior, which may indirectly disturb feeding behavior

(Supplementary Fig. 4k-l, Supplementary Movie 4). The DMH was previously reported to be involved in behavioral thermoregulation[39]. Therefore, the regulatory effect of the apMPOA→DMH pathway on feeding behavior awaits further verification. In the controls, light delivery to the apMPOA→ARC/PVH/DMH terminal fibers with YFP expression did not result in changes of food intake or rectal temperature (Supplementary Fig. 4f–h, j). Additionally, terminal activations on the other downstream targets, including LH, BNST, PAG, PVT, TN, BA, and VTA, had no effect on food intake or rectal temperature (Supplementary Fig. 4c–e, i).

**Caspase ablation and chemogenetic inhibition on the post-synaptic neurons of apMPOA to ARC/PVH pathways reversely modulates feeding behavior.** Considering that optogenetic terminal activation might cause nonspecific antidromic stimulation of collateral targets, we next assessed the necessity of the apMPOA→ARC/PVH/DMH pathways for regulating feeding behavior by injecting AAV1-cre-GFP into the apMPOA and AAV-flex-taCasp3-TEVp into the ARC, PVH, and DMH (Fig. 4a). With this approach, the apMPOA-recipient post-synaptic neurons in the ARC, PVH, and DMH would be selectively ablated after expressing Caspase-3. In the controls, neurons express GFP but not Caspase-3 will not be ablated (Fig. 4b, d and

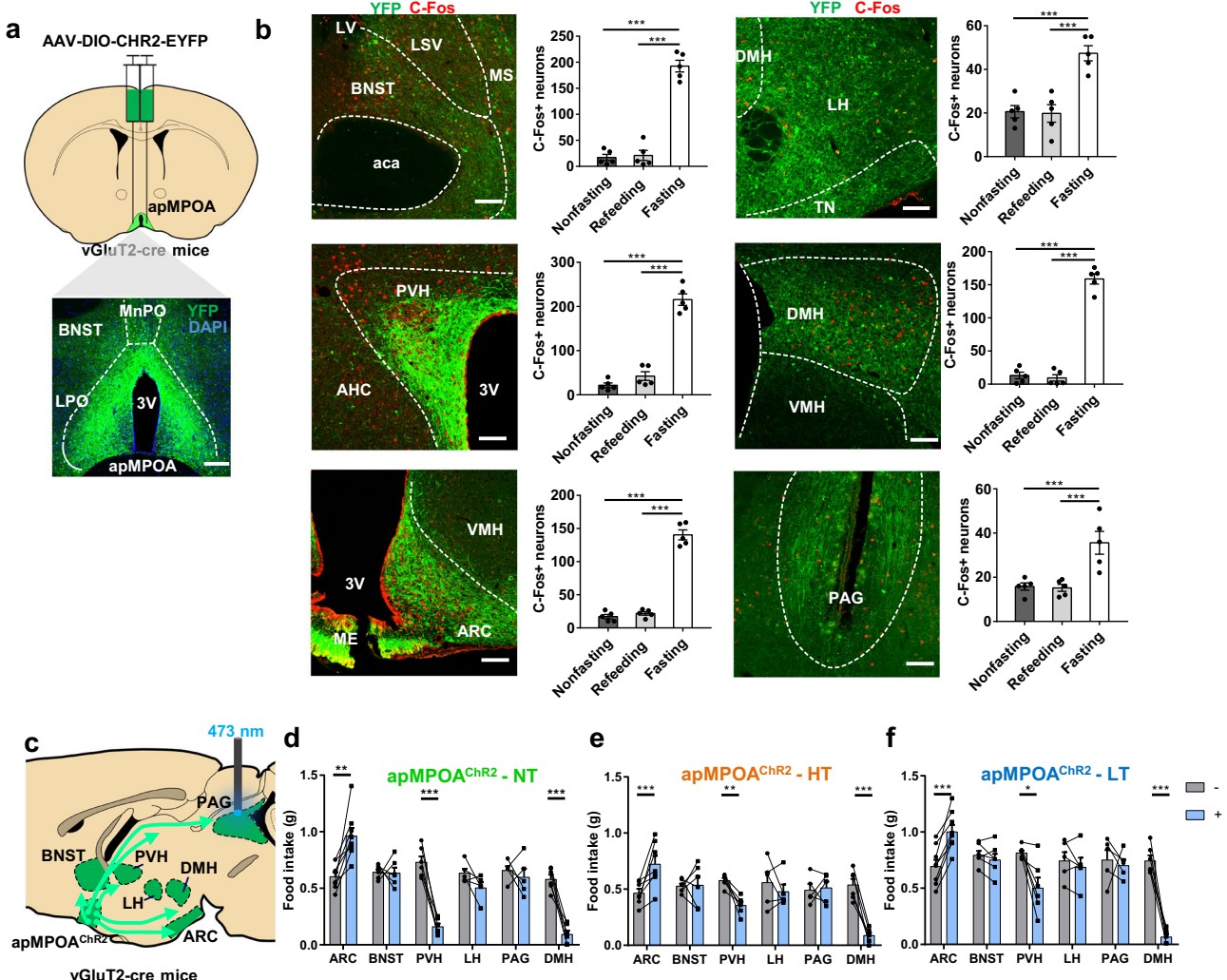

**Fig. 3 Optogenetic terminal activations on the ARC and PVH orchestrate food intake. a** Schematic and representative coronal images of injecting AAV virus expressing ChR2 into the apMPOA in vGluT2-cre mice. Image representative of $n = 5$ mice for each group of downstream targets. Scale bar, 100 µm. **b** Representative images of downstream targets of glutamatergic apMPOA neurons (green, ChR2), including BNST, LH, PVH, DMH, ARC and PAG, in which fasting-induced C-Fos expression can be clearly observed compared to the re-feeding and no fasting groups (red, C-Fos). Scale bar, 100 µm. One-way ANOVA and post hoc LSD multiple comparison, ***$p < 0.001$, $n = 5$ animals for each group. **c** Schematic sagittal image of terminal activation on individual downstream target of apMPOA (PAG shown here). **d–f** Two-hour food intake with (+, blue boxes) and without (-, gray boxes) opto-stimulation on the target regions during ambient temperatures. The different ambient temperatures were applied on the same animals with 3–5 days interval. Two-way repeated-measure ANOVA and post hoc LSD multiple comparisons were performed on the food intake of mice with each target region activated. Optogenetic activation on the ARC, PVH and DMH caused significant changes of food intake compared with no stimulation. ARC: Interaction effect: $F_{2,11}$(temperature × treatment) = 1.19, $p = 0.34$; main effect: $F_{2,11}$(temperature) = 7.2, **$p = 0.01$; $F_{1,12}$(treatment) = 23.55, ***$p < 0.001$. Pairwise comparisons showed that optogenetic activating on ARC increased food intake in all thermal conditions (***$p < 0.001$, **$p = 0.003$). $n = 8$. PVH: Interaction effect: $F_{2,7}$(temperature × treatment) = 6.32, *$p = 0.027$; main effect: $F_{2,7}$(temperature) = 10.17, **$p = 0.008$; $F_{1,8}$(treatment) = 40.11, ***$p < 0.001$. Pairwise comparisons showed that optogenetic activating on PVH decreased food intake in all thermal conditions (***$p < 0.001$, **$p = 0.002$, *$p = 0.04$). $n = 6$. DMH: Interaction effect: $F_{2,9}$(temperature × treatment) = 6.19, *$p = 0.02$; main effect: $F_{2,9}$(temperature) = 10.23, **$p = 0.005$; $F_{1,10}$(treatment) = 197.4, ***$p < 0.001$. Pairwise comparisons showed that optogenetic activating on DMH decreased food intake in all thermal conditions (***$p < 0.001$). $n = 6$. All error bars show SEM.

Supplementary Fig. 7b). The ablation of apMPOA-recipient ARC neurons reduced food intake during normothermia by 32.9% (Fig. 4e). Reduced food intake was also observed at high and low ambient temperature (Fig. 4e). In contrast, the ablation of apMPOA-recipient PVH neurons increased food intake during normothermia by 62.3%, as well as high and low ambient temperature (Fig. 4c). The ablations of both pathways specifically mediate food intake, as they had no effects on rectal temperature (Supplementary Fig. 7d-e). Regarding the apMPOA→DMH pathway, we selectively ablated apMPOA-recipient DMH

neurons (Supplementary Fig. 7b), and found that the ablation had no influence on food intake during normothermia (Supplementary Fig. 7c). This indicated that the apMPOA→DMH pathway did not contribute to feeding regulation. Moreover, the ablation of apMPOA→DMH pathway resulted in defective thermoregulation under rapid thermal challenge, since their rectal temperature fluctuated as ambient temperature changed (Supplementary Fig. 7f). The thermal imbalance might indirectly affect food intake during both high and low ambient temperatures (Supplementary Fig. 7c). These results indicated that the

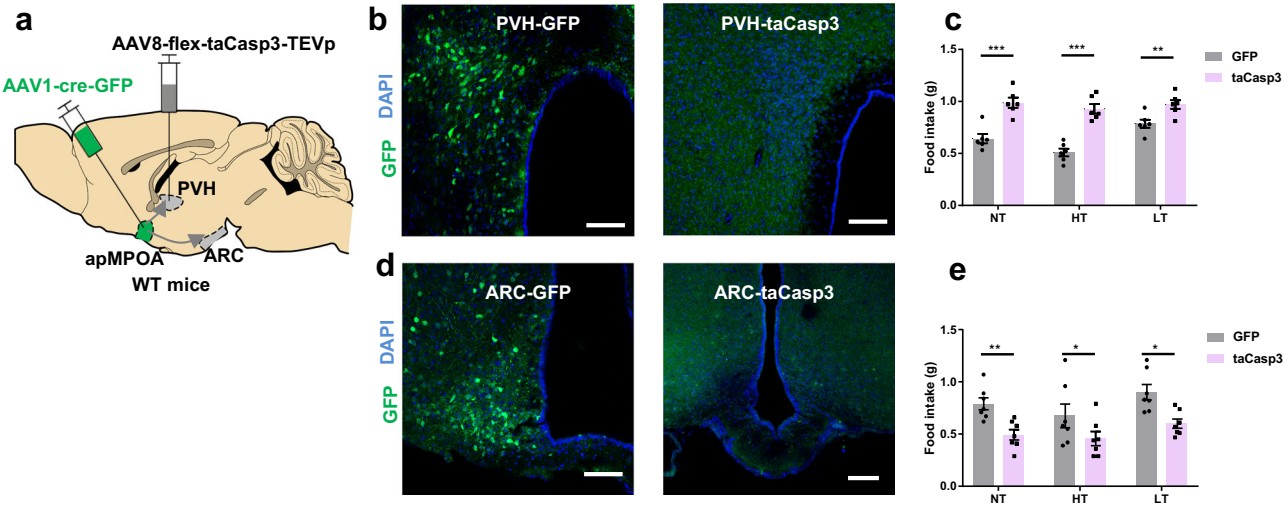

**Fig. 4 Genetic ablation on the apMPOA-recipient neurons in ARC/PVH reversely modulate feeding behavior. a** Schematic image showing genetic ablation of apMPOA-recipient neurons in ARC/PVH individually (PVH shown here) in wild type mice by injecting anterograde transsynaptic AAV1-Cre-GFP in apMPOA, and Cre-dependent caspase-3 in the ARC and PVH. **b, d** Representative images showing preserved transsynaptic neurons in PVH (upper left image) and ARC (lower left image) that expressed GFP in the control group, but ablated transsynaptic neurons in the PVH (upper right image) and ARC (lower right image) in taCasp3-injected mice. Scale bar, 100 μm. **c** The ablation of transsynaptic neurons in the PVH significantly increased food intake compared to that in the controls in all thermal conditions. Interaction effect: $F_{2,9}$ (temperature × treatment) =10.67, **$p = 0.004$; main effect: $F_{2,9}$(temperature) = 25.85, ***$p < 0.001$; $F_{1,10}$(treatment) = 42.42, ***$p < 0.001$. Pairwise comparisons showed that caspase ablating apMPOA-recipient PVH neurons increased food intake in all thermal conditions (***$p < 0.001$, **$p = 0.01$). Within-group tests showed significant simple effect of temperature in control group ($F(2,9) = 34.52$,***$p < 0.001$), but not in the taCasp3 group ($F(2,9) = 2.0$, $p = 0.191$). Two-way repeated-measure ANOVA and post hoc LSD multiple comparison, $n = 6$ animals for each group. **e** The ablation of transsynaptic neurons in the ARC significantly decreased food intake compared to that in the controls in all thermal conditions. Interaction effect: $F_{2,11}$(temperature × treatment) = 0.63, $p = 0.55$; main effect: $F_{2,11}$(temperature) = 16.4, ***$p = 0.001$; $F_{1,12}$(treatment) = 9.35, **$p = 0.01$. Pairwise comparisons showed that caspase ablating apMPOA-recipient ARC neurons decreased food intake in all thermal conditions (***$p < 0.001$, **$p = 0.005$). Within-group tests showed significant simple effect of temperature in the control group ($F(2,11) = 11.39$,**$p = 0.002$), and the taCasp3 group ($F(2,11) = 5.64$, *$p = 0.021$). Two-way repeated-measure ANOVA and post hoc LSD multiple comparison, $n = 7$ animals for each group. All error bars show SEM.

apMPOA→DMH pathway differs from the other two effector pathways, and may be primarily involved in autonomic and behavioral thermoregulation. Thus, the role of this pathway in the direct regulation on feeding behavior was excluded.

We also reversely manipulated the apMPOA circuits using chemogenetic inhibition on the apMPOA-recipient neurons in the ARC and PVH that express inhibitory designer receptor exclusively activated by designer drugs (DREADD-Gi), hM4Di, and could be inhibited after intraperitoneal injection of clozapine-N-oxide (CNO) (Fig. 5a, b, e). The mice with DREAD-Gi expressed in apMPOA-recipient ARC neurons showed lower food intake after CNO injection (1.2 mg/kg) at all three ambient temperatures (Fig. 5f). Whereas the chemogenetic inhibition on the postsynaptic PVH neurons induced increased food intake after CNO injection (1.2 mg/kg) (Fig. 5c). No effect of CNO on food intake was observed in control mice with mCherry expressed in the postsynaptic neurons in ARC and PVH (Fig. 5d, g). Consistent with the optogenetic activation and cell ablation, chemogenetic inhibition on the apMPOA→ARC and PVH pathways further validated their regulatory roles on feeding behavior.

It is worth mentioning that there were significant interaction effects between temperature and treatment in the experiments of caspase ablating or chemogenetic inhibiting apMPOA-recipient PVH neurons (Figs. 4c and 5c). Within-group comparisons on food intake of the mice with apMPOA-recipient PVH neurons genetically manipulated among the three ambient temperatures (purple and red boxes in Figs. 4c and 5c), showed absent effect of temperature on food intake, suggesting that the ablation or inhibition of the apMPOA→PVH pathway reversely eliminated the temperature dependence of feeding behavior as observed in

the control groups (e.g., LT > NT > HT patterns depicted as gray boxes of the controls in Figs. 4c and 5c). This finding further proves that apMPOA→PVH projections play a vital role in the ambient temperature dependence of feeding behavior, whereas cutting off apMPOA→PVH pathway result in the loss of this property.

**Thermosensory characteristics of ARC and PVH-projecting neurons in the apMPOA.** The manipulations outlined above demonstrate that the apMPOA→ARC/PVH pathways orchestrate feeding behavior. However, it remains unclear whether the apMPOA-innervated circuits regulate feeding behavior by incorporating ambient thermal inputs. Here, we investigated the thermosensory characteristics of ARC/PVH-projecting apMPOA neurons using retrograde tracing and optical photometry. We injected AAVretro-DIO-Flp (mixed with red fluorescent biotinylated dextran amines to indicate the injection range) into the ARC/PVH and Flp-dependent AAV-fDIO-GCaMP6s into the apMPOA of vGluT2-cre mice, and installed an optical fiber above the apMPOA to record calcium responses to two types of stimulations: local stimulation on a thermal plate and ambient exposure in an environmental chamber (Fig. 6a, b). We discovered that the ARC-projecting apMPOA neurons exhibited no calcium responses to local warming ($25 \rightarrow 40\,°C$) or cooling ($25 \rightarrow 10\,°C$) on the thermal plate compared with mice subjected to neutral stimulation ($25\,°C$, upper parts in Fig. 6c–f and Supplementary Fig. 8b–d). The same mice showed obviously elevated calcium activity in a cold chamber ($25 \rightarrow 10\,°C$), but no responses in a hot chamber ($25 \rightarrow 35\,°C$, upper parts in Fig. 6h–k and Supplementary Fig. 8f–h). The PVH-projecting apMPOA

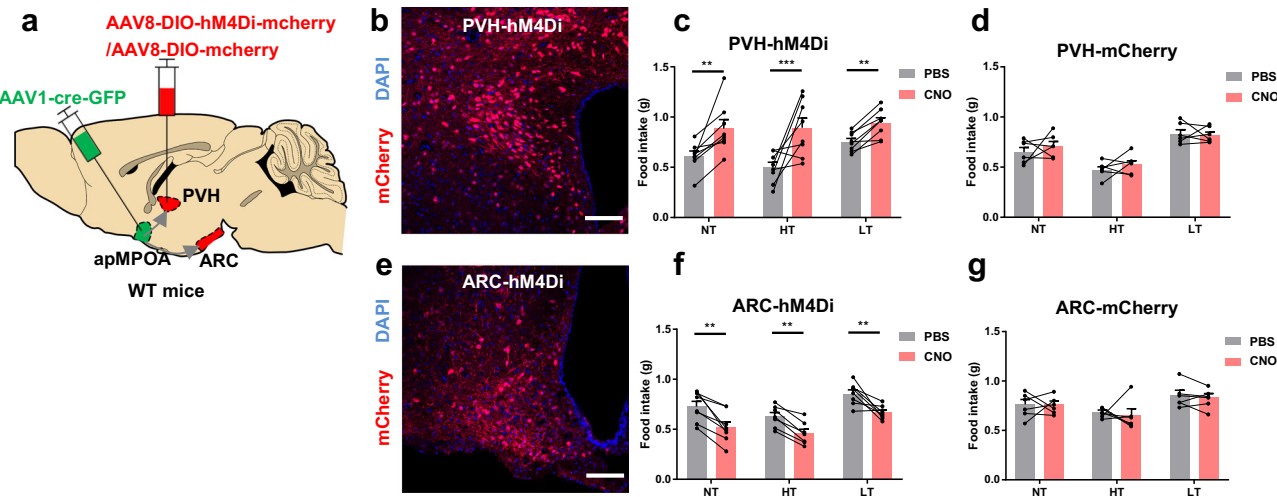

**Fig. 5 Feeding behavior after chemogenetic inhibition on the apMPOA-recipient neurons in ARC/PVH. a** Chemogenetic inhibiting apMPOA-recipient neurons in ARC/PVH individually in wild type mice by injecting AAV1-Cre-GFP in apMPOA and DREAD-Gi in the ARC and PVH. Transsynaptic neurons in the PVH (**b**) and ARC (**e**) expressed hM4Di-mCherry. Image representative of $n = 8$ mice. Scale bar, 100 μm. **c** Chemogenetic inhibiting postsynaptic PVH neurons significantly increased food intake after CNO injection under all thermal conditions compared to PBS. Interaction effect: $F_{2,13}$(temperature × treatment) = 6.56, *$p = 0.014$; main effect: $F_{2,13}$(temperature) = 4.95, *$p = 0.026$; $F_{1,14}$(treatment) = 56.82, ***$p < 0.001$. Pairwise comparisons showed that chemogenetic inhibiting apMPOA-recipient PVH neurons increased food intake in all thermal conditions (***$p < 0.001$, **$p < 0.01$, *$p < 0.05$). Within-group tests showed significant simple effect of temperature in control group (F(2,13) = 5.08, *$p = 0.023$), but not in the taCasp3 group (F(2,13) = 2.42, $p = 0.13$). Two-way repeated-measure ANOVA and post hoc LSD multiple comparison, $n = 8$ animals. **f** Chemogenetic inhibiting postsynaptic ARC neurons significantly decreased food intake. Interaction effect: $F_{2,11}$(temperature × treatment) = 0.46, $p = 0.64$; main effect: $F_{2,11}$(temperature) = 8.81, **$p = 0.006$; $F_{1,12}$(treatment) = 11.78, **$p = 0.005$. Pairwise comparisons showed that chemogenetic inhibiting apMPOA-recipient ARC neurons decreased food intake in all thermal conditions (all **$p < 0.01$). Within-group tests showed significant simple effect of temperature in the control group (F(2,11) = 4.82, *$p = 0.031$), and the taCasp3 group (F(2,11) = 5.13, *$p = 0.027$). Two-way repeated-measure ANOVA and post hoc LSD multiple comparison, $n = 8$ animals. **d, g** The controlled groups that expressed mCherry did not show food intake changes between CNO and PBS injections. PVH-mCherry (**d**): interaction effect: $F_{2,9}$(temperature × treatment) = 0.47, $p = 0.64$; main effect: $F_{2,9}$(temperature) = 35.98, ***$p < 0.001$: $F_{1,12}$(treatment) = 1.55, $p = 0.24$. ARC-mCherry (**g**): interaction effect: $F_{2,9}$(temperature × treatment) = 1.5, $p = 0.28$; main effect: $F_{2,9}$(temperature) = 18.15, ***$p = 0.001$; $F_{1,10}$(treatment) = 1.97, $p = 0.19$. Two-way repeated-measure ANOVA and post hoc LSD multiple comparison, $n = 6$ animals for each group. All error bars show SEM.

neurons showed no calcium response on the warm plate, but relatively reduced one on the cold plate (lower parts of Fig. 6c–e, g and Supplementary Fig. 8b–d). The same mice showed increased calcium activity in a hot chamber, but reduced one in a cold chamber (lower parts in Fig. 6h–j, l and Supplementary Fig. 8f-h). These findings suggested that ARC-projecting apM-POA neurons were selectively activated by cooling, and PVH-projecting neurons had bidirectional thermosensory characteristics. Importantly, the PVH and ARC-projecting apMPOA neurons specifically responded to ambient thermal exposure, rather than local thermal stimuli (PVH-projecting neurons showed weak responses to local cooling). These findings were further validated by colocalizations between thermal-induced C-Fos and retrograde traced apMPOA neurons (Supplementary Fig. 9). The distinct thermosensory characteristics of the two subsets of apMPOA neurons may be the neural basis of ambient temperature dependence of feeding behavior. In sensing external thermal inputs, the apMPOA neurons with distinct thermosensory characteristics orchestrate feeding behavior through fiber projections to anorexigenic or orexigenic neurons that are densely distributed in the PVH and ARC.

**Anatomical and molecular differentiations of the two populations of apMPOA neurons using projection-specific RNA sequencing.** We next sought to explore the evidence for the heterogeneity of the ARC and PVH-projecting neurons in the apMPOA. We first examined their anatomical colocalizations by injecting CTB-555 and CTB-488 into the PVH and ARC respectively (Fig. 7a). The retrograde-labeled neurons from PVH

and ARC were sparsely colocalized with each other (approximately 9% of CTB-488+ neurons expressed CTB-555, whereas 14% of CTB-555+ neurons expressed CTB-488), indicating that the ARC and PVH-projecting neurons in the apMPOA were two neighboring but separate populations (Fig. 7b, c). This finding was also identified by cell-type specific retrograde tracing for ARC and PVH-projecting apMPOA neurons in the vGluT2-cre and GAD2-cre mice (Supplementary Fig. 10).

Given the distinct characteristics in sensing ambient temperatures (Fig. 6h–j), we next investigated their neurotransmission inputs of two neuronal populations using retrograde rabies virus (RV). We injected Cre-dependent AAVretro vectors expressing EGFP-TVA and RG into the PVH or ARC (not in the same mouse) of vGluT2-cre mice. After allowing three weeks for expression in the retrograde-labeled neurons in apMPOA, we injected modified rabies virus RV-EnvA-ΔG-dsRed into the apMPOA (Fig. 7d). The control experiments for RV were performed by injecting RG and RV-EnvA-ΔG-dsRed into vGluT2-cre mice without TVA. In addition, all viruses were injected in wild-type mice to exclude potential leaky expression (Supplementary Fig. 12). With this viral strategy, the RV could specifically label the anatomical inputs to apMPOA neurons that projecting to PVH and ARC. Both PVH and ARC-projecting apMPOA neurons co-received monosynaptic inputs from multiple upstream regions (Supplementary Fig. 11). Importantly, their monosynaptic inputs showed a certain distinction. Specifically, strong RV labeling was observed in the posteromedial part of the BNST (pBNST), dorsal subnuclei of lateral parabrachial nucleus (LPBd), nucleus tractus solitaries (NTS), and BA to the PVH-projecting apMPOA neurons (Fig. 7e, f). In contrast, the

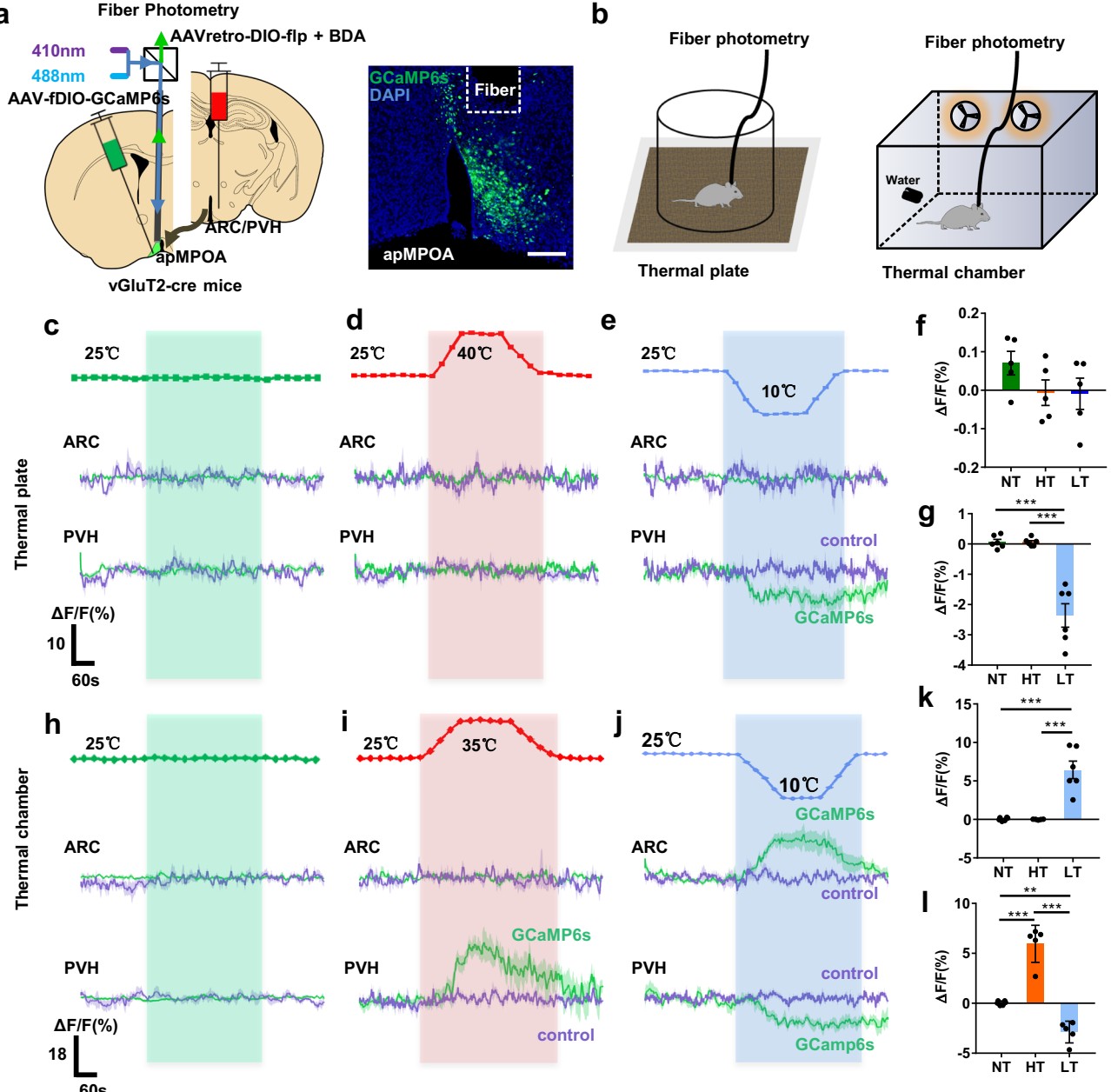

**Fig. 6 PVH and ARC-projecting apMPOA neurons are responsive to ambient thermal exposure, but not to local stimuli. a** Schematic and representative coronal images of AAVretro-DIO-flp virus injection into the PVH and ARC (ARC shown here) and AAV-fDIO-GCaMP6s into the apMPOA in vGluT2-cre mice. The optical fiber was placed 200 μm above the apMPOA neurons. Red fluorescent BDA (biotinylated dextran amines) was mixed into the virus to indicate the injection range. Image representative of n = 12 mice. Scale bar, 100 μm. **b** Schematic of thermal exposure by temperature-controlled plate and chamber. **c–e** GcaMP6s (mean, green line; SEM, green shading) and control (mean, purple line; SEM, purple shading) signal changes (ΔF/F%) of the ARC (top) and PVH-projecting (bottom) apMPOA neurons in response to local thermal stimuli: neutral (25 °C, green dotted line in Fig. 6c), hot (25 °C → 40 °C, red dotted line in Fig. 6d), cold (25 °C → 10 °C, blue dotted line in Fig. 6e). **h–j** GcaMP6s (mean, green line; SEM, green shading) and control (mean, purple line; SEM, purple shading) signal changes (ΔF/F%) of the ARC (top) and PVH-projecting (bottom) apMPOA neurons in response to ambient thermal exposure: neutral (25 °C, green dotted line in Fig. 6h), hot (25 °C → 35 °C, red dotted line in Fig. 6i), cold (25 °C → 10 °C, blue dotted line in Fig. 6j). Bar graphs show mean ΔF/F response of ARC (thermal plate: **f** thermal chamber: **k**) and PVH-projecting (thermal plate: **g** thermal chamber: **l**) apMPOA neurons in a 5 min window after the onset of thermal exposure. One-way repeated-measure ANOVA and post hoc LSD multiple comparison, **g** all ***$p < 0.001$, **k** all ***$p < 0.001$, **l**: **$p = 0.002$, ***$p < 0.001$. $n = 6$ for both PVH and ARC groups, but $n = 5$ animals were included into statistical ANOVA in ARC groups in cold plate and hot chamber, PVH group in cold chamber and hot chamber due to incomplete data. All error bars show SEM.

ARC-projecting apMPOA neurons received sparse inputs from these regions, but abundant inputs from dense cell bodies in the external subnuclei of LPB (LPBe, Fig. 7e, g). Among these upstream regions, the LPB has been previously identified to transmit cutaneous thermosensory signals of ambient temperature, with the LPBd transmitting warming signals to glutamatergic MPOA neurons, whereas the LPBe transmitting cutaneous cooling input to GABAergic neurons in MnPO, thus facilitating thermoregulation[40,41]. In addition to the LPB, we also found strong RV labeling in the pBNST, NTS, and BA, which are

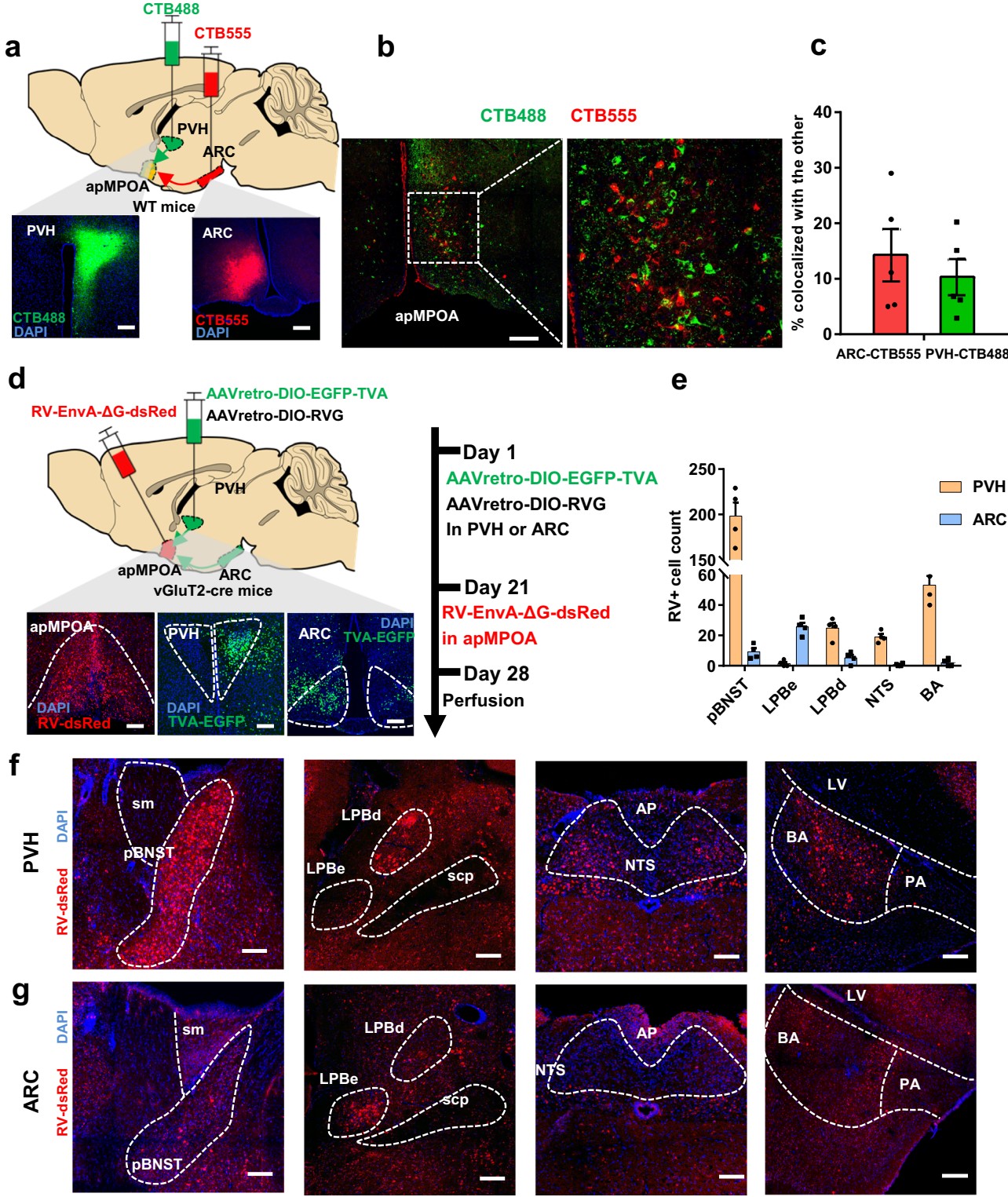

not considered in the current study because they were not previously reported to be involved in thermosensation or transmission[42,43].

To molecularly discriminate the two subsets of ARC/PVH-projecting apMPOA neurons, we collected them in acute sections with a glass pipette after retrograde labeling and sequenced the RNA using SmartSeq2 protocol[44] (Fig. 8a). We injected a Cre-dependent AAVretro vector expressing EGFP into the PVH or ARC (not in the same mouse) of vGluT2-cre mice (Fig. 8a). After allowing three weeks for the expression of EGFP, we dissected

acute brain slices containing apMPOA and collected fluorescently labeled cells in the apMPOA with glass pipettes (Fig. 8a). A total of 5 biological repeats were collected for the PVH and ARC groups, respectively, with 15 cells for one mouse sample. A total of 15,682 and 15,889 transcripts were expressed in PVH and ARC-projecting apMPOA neurons, with 765 and 972 transcripts being uniquely expressed in each group and 14917 transcripts were co-expressed in both group (Fig. 8b and Supplementary Table 1). A total of 455 significant differentially expressed genes (DEGs) were found, with 221 DEGs upregulated in

**Fig. 7 Retrograde labeling for the two subsets of anatomically differentiated PVH and ARC-projecting apMPOA neurons. a** Schematic and representative coronal images of the injection of CTB 488 and CTB 555 into PVH and ARC of wild type mice respectively. Image representative of $n = 5$ mice. Scale bar, 100 μm. **b**, **c** Colocalizations of retrograde-labeled neurons in the apMPOA from the PVH (green) and ARC (red), $n = 5$ animals for each group (a total of 8 mice models were made with 3 mice excluded due to missed injection). Scale bar, 100 μm. **d** Schematic and representative coronal images of the injection of Cre-dependent AAVretro vector expressing RV helpers into the PVH or ARC on day 1 and RV-EnvA-ΔG-dsRed on day 21 into the apMPOA of vGluT2-cre mice. Image representative of $n = 4$ mice. Scale bar, 100 μm. **e** Quantification of numbers of retrogradely labeled cells in different regions in ipsilateral sides of the injected apMPOA. $n = 4$ animals for each group (a total of 6 mice models were made with 2 mice excluded due to missed injection). **f** Typical coronal images of DsRed-expressing neurons in posterior part of the BNST (pBNST), LPBd, NTS, and BA in the group with RV helpers-injection into the PVH. Image representative of $n = 4$ mice. Scale bar, 100 μm. **g** Typical coronal images of DsRed-expressing neurons in LPBe, but sparse neurons in the pBNST, NTS and BA in the group with RV helpers-injection into the ARC. Image representative of $n = 4$ mice. Scale bar, 100 μm. All error bars show SEM.

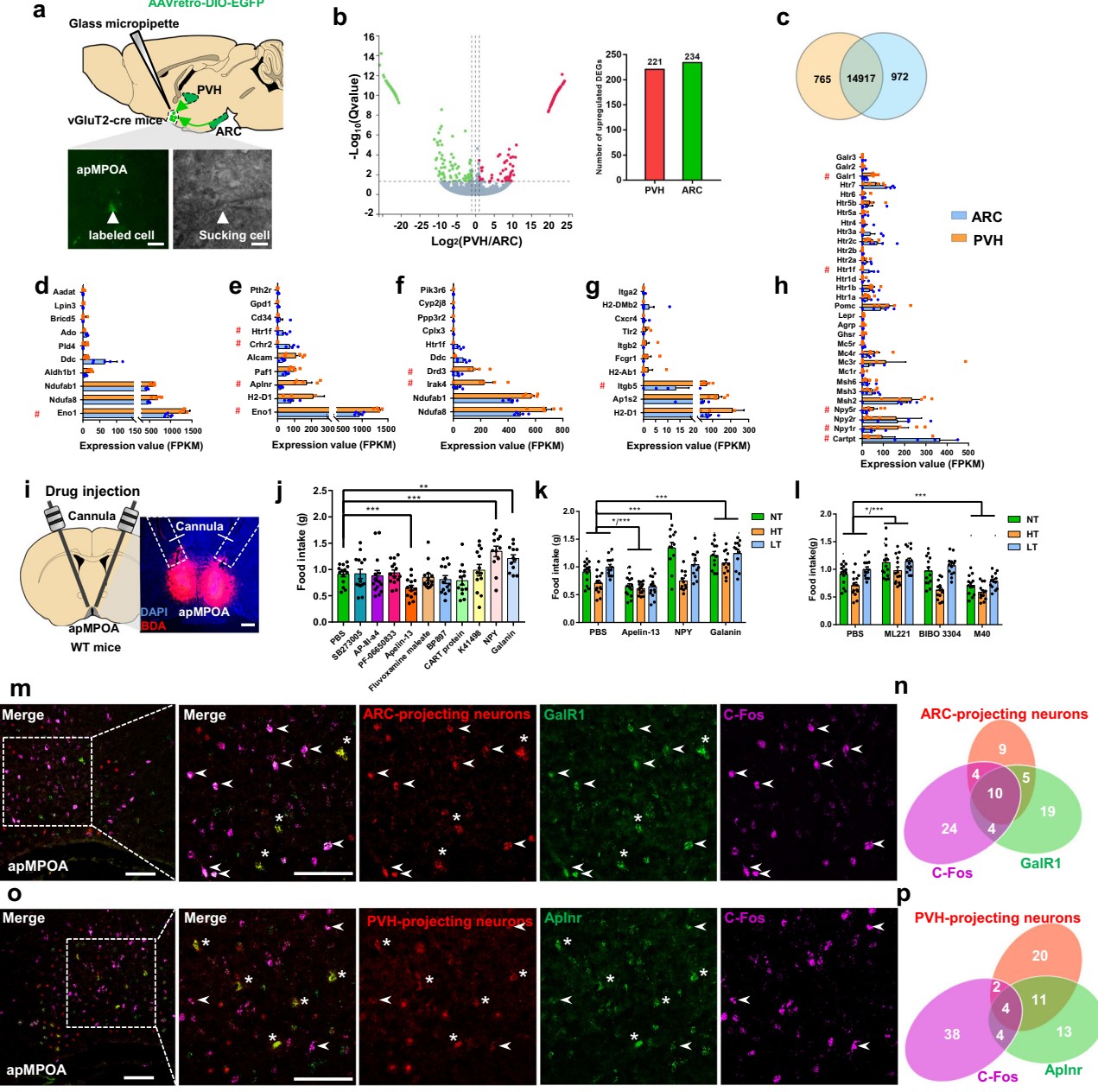

PVH-projecting apMPOA neurons and 234 DEGs upregulated in ARC-projecting neurons (Fig. 8b, c and Supplementary Table 2). We performed gene annotation analysis via KEGG pathway category on the generated 455 DEGs, that were divided into categories including metabolism, environmental information processing, organismal systems, cellular processes. We investigated potential candidate genes among the top ten expressed DEGs in each category (Fig. 8d–g). We focused on a set of DEGs depending on their functions in metabolism-related cellular processing and intercellular transmission activity, including the

**Fig. 8 Molecular markers for the PVH and ARC-projecting apMPOA neurons. a** Schematic image of injecting AAVretro-GFP into PVH or ARC, and GFP-labeled cell body sucked by a micropipette. Scale bar, 20 μm. **b** Volcano plot and bar graph depicting the transcript abundance between PVH-projecting (red) and ARC-projecting (green) apMPOA neurons. **c** Venn diagram depicting the shared and unique gene expressions of the two neuronal populations. The expression (FPKM) of top ten DEGs in the KEGG pathway categories: metabolism (**d**), environmental information processing (**e**), organismal systems (**f**), cellular processes (**g**). The candidate marker genes are indicated in red #. $n = 5$ animals for each group. **h** The genes related to feeding peptides and receptors. The candidate marker genes are indicated in red #. $n = 5$ animals for each group. **i** Schematic and representative images of cannula injection of antagonists or agonists of the candidate genes. Scale bar, 100 μm. **j** Injection of antagonists or agonists of the selected candidate genes during normothermia shows that apelin, galanin, and NPY injections cause food intake changes. One-way repeated-measure ANOVA and post hoc LSD multiple comparison, ***$p < 0.001$, **$p = 0.01$. $n = 14$ animals. **k** Feeding test after apelin, galanin and NPY injections. Compared with PBS, apelin-13 robustly suppressed food intake, and the galanin robustly promoted food intake at ambient temperatures. NPY only promoted food intake during normothermia, but not during high or low temperature. Two-way repeated-measure ANOVA and post hoc LSD multiple comparison. Interaction effect: $F_{6,96}$(temperature × drug) = 6.03, ***$p < 0.001$; main effect: $F_{2,47}$(temperature) = 29.81, ***$p < 0.001$; $F_{3,48}$(drug) = 28.62, ***$p < 0.001$. $n = 14$ animals. **l** Feeding test after apelin, galanin, and NPY receptors antagonists injections. Compared with PBS, ML221 (apelin receptor antagonist) and M40 (GalR1 antagonist) reversed the food intake by the agonists. Blocking the NPY receptor (BIBO 3304) did not induce food intake changes. Two-way repeated-measure ANOVA and post hoc LSD multiple comparison. Interaction effect: $F_{6,100}$(temperature × drug) = 2.4, *$p = 0.033$; main effect: $F_{2,49}$(temperature) = 29.81, ***$p < 0.001$; $F_{3,50}$(drug) = 13.43, ***$p < 0.001$. $n = 14$ animals. **m** Colocalizations of *GalR1* (green), ARC-projecting apMPOA neurons (red), and fasting-induced C-Fos (purple). Image representative of n = 5 mice. Scale bar, 100 μm. Arrowhead denotes *GalR1*+, ARC-projecting+, and C-Fos+ neurons, asterisk denotes *GalR1*+ and ARC-projecting+ neurons. **n** ARC-projecting neurons presented colocalization with fasting-induced C-Fos, and galanin receptor 1.
**o** Colocalizations of *Aplnr* (green), PVH-projecting apMPOA neurons (red), and fasting-induced C-Fos (purple). Image representative of $n = 5$ mice. Scale bar, 100 μm. Arrowhead denotes *Aplnr*+, PVH-projecting+, and C-Fos+ neurons, asterisk denotes *Aplnr*+ and PVH-projecting+ neurons. **p** PVH-projecting neurons presented colocalizations with apelin receptor, but rarely with fasting-induced C-Fos. All error bars show SEM.

genes *Eno1*, *Aplnr*, *Crhr2*, *Htr1f*, *Drd3*, *Irak4*, and *Itgb5* (Fig. 8d–g). Besides data-driven gene annotation analysis, we also analyzed a range of genes related to appetite neuropeptides or receptors which might not fall into the top ten of gene annotation analysis above (Fig. 8h), and found that the genes *Htr1f*, *NPY1r*, *GalR1*, *Cartpt*, and *NPY5r* were group-differed. Collectively, we selected the DEGs *Htr1f*, *Irak4*, *Aplnr*, *Itgb5*, *Eno1*, *Crhr2*, *Drd3*, *GalR1*, *Cartpt*, *Npy1r*, and *Npy5r* as candidate marker genes for further behavioral verification (red # in Fig. 8d–h).

We tested behavioral performances of mice after microinjecting inhibitors or agonists of these candidate genes into apMPOA (Fig. 8i). We found that injection of apelin in the apMPOA suppressed food intake of starved mice during normothermia (Fig. 8j). The inhibiting effect of apelin in the apMPOA was robust in response to ambient temperatures (Fig. 8k). Blocking the apelin receptors in the apMPOA during ambient temperature robustly promoted food intake compared with the results of PBS injection (Fig. 8l). The modulatory effect of pharmacological apelin application on feeding behavior coincided with optogenetic/chemogenetic/caspase ablating manipulations on apMPOA→PVH pathway (Figs. 3d–f, 4c, and 5c) further indicating that the *Aplnr* might be a potential marker for the PVH-projecting apMPOA neurons. In contrast to the apelin, microinjection of galanin in the apMPOA also increased food intake during normothermia (Fig. 8j). The promoting effect of galanin in the apMPOA was also robust at high and low temperatures (Fig. 8k). Conversely, blocking the galanin receptor 1 in the apMPOA reversed the promoting effects of galanin neurons on food intake in response to ambient temperatures (Fig. 8l), suggesting that the *GalR1* might be a potential marker for the ARC-projecting apMPOA neurons. Additionally, NPY (neuropeptide Y) injection showed promoting effect during normothermia (Fig. 8j), but not during high or low ambient temperature (Fig. 8k), and blocking the NPY1 receptors in the apMPOA had no effect on feeding (Fig. 8l).

In addition of pharmacological verification, we further performed fluorescence in situ hybridization (FISH) by using RNAscope to verify the expressions of *Aplnr* and *GalR1* in PVH and ARC-projecting apMPOA neurons that got C-Fos activated by fasting. We injected Cre-dependent AAVretro vectors expressing mCherry into the PVH or ARC (not in the same mouse) of vGluT2-cre mice, and performed FISH for anti-apelin

receptor and anti-galanin receptor 1 and C-Fos three weeks later. Specifically, The ARC-projecting neurons in the apMPOA exhibited substantial colocalization with GalR1. Approximately 53% of ARC-projecting neurons expressed galanin receptor 1 (Fig. 8m, n), but minor colocalization with apelin receptor (approximately 6.9%, Supplementary Fig. 13a, b). Approximately 40.5% of the PVH-projecting apMPOA neurons were overlapped with apelin receptor (Fig. 8o, p), but rarely with galanin receptor 1 (approximately 20%, Supplementary Fig. 13c, d). These findings indicated that the two subsets of ARC-projecting and PVH-projecting apMPOA neurons were molecularly distinct. More ARC-projecting neurons (approximately 42.5%, Fig. 8m, n and Supplementary Fig. 13a, b) than PVH-projecting labeled neurons (approximately 12.5%, Fig. 8o, p and Supplementary Fig. 13c, d) were colocalized with the fasting-induced C-Fos, suggesting that hunger-related neurons in apMPOA are highly overlapped with ARC-projecting neurons.

## Discussion

Here, we describe previously unrecognized neuronal populations and circuits for coordinating feeding behavior in coping with ambient temperature. The regulatory effect is centrally orchestrated by two subsets of glutamatergic neurons in the apMPOA projecting to the ARC and PVH. One subset of apMPOA neurons shows an active response to low ambient temperature, enacting a promoting food intake through terminal fibers projecting to the ARC. Conversely, the other subset of glutamatergic neurons, which project to the PVH, inhibits food intake, showing an active response to high ambient temperature. By receiving heterogeneous anatomical inputs from different parts of LPB, the two subsets of apMPOA neurons, which are molecularly distinct in terms of apelin and galanin receptor occurrence, adaptively mediate feeding behavior in coping with ambient thermal challenges.

**Feeding behavior is modulated by the demand of thermal homeostasis.** Feeding is a complex innate behavior that maintains energy homeostasis and is regulated by a variety of hard-wired neural circuits[45]. The survival-relevant innate behaviors such as hunger, thirst, pain, fear, and anxiety, are hierarchically regulated[46–48]. The most urgent environmental or physiological

factors are prioritized when they threaten survival[49]. In the instances of thermal challenges, thermal homeostasis tends to be the highest priority process for ensuring survival by all possible means of regulation, such as BAT thermogenesis, shivering, and vascular contraction and dilation[50], regardless of hunger or satiety. As a key means of behavioral thermoregulation[51], the adaptive feeding behavior would help the body defend against extreme thermal challenges. Promoted feeding would provide a greater energy supply in response to cold environments, conversely, the inhibition of feeding would impede extra heat production in a hot environment. From this perspective, we reason that feeding behavior acts as an indispensable mechanism of thermoregulation at extreme ambient temperatures, in which the body regulates the amount of food intake according to the demand for heat production and dissipation, rather than glucostasis or lipostasis[52,53]. The potential contribution of feeding to thermoregulation is easily overlooked under normothermic conditions. However, it cannot be ignored that the coordinating regulatory effect of ambient temperature on feeding is a common phenomenon in mammals at extreme temperatures[9]. Likewise, feeding in humans shows a certain dependence on temperature, and climate, especially for soldiers in extreme environments[10–12].

**Neural circuits underlying ambient temperature-regulated feeding behavior.** The apMPOA contains diverse cell types[54], and intricate neural circuits that innervate a series of downstream homeostatic-related targets[8,17,34,55]. Supporting this, optogenetic activation on apMPOA neurons induced both temperature and feeding behavior alterations (Fig. 2). Our study further broadened the regulatory role of apMPOA in orchestrating temperature-dependent feeding behavior. Overriding energy demand, feeding behavior acts as a mechanism for thermoregulation likely requiring hard-wired neural circuits that intertwine temperature and food signals.

The unique anatomical downstream and upstream targets of apMPOA facilitate rapid sensing and integrating ambient temperature and energy state, and exert a corresponding regulatory role. Anatomically, the apMPOA is located along the third ventricle and extends dense fibers to nearby circumventricular nuclei involved in energy homeostasis, such as DMH, LH, PVH, ARC[30,31]. We infer that the essential effectors of feeding behavior and autonomic thermoregulation co-orchestrated by apMPOA lie in the distinct downstream regions. Noticeably, the ablations of postsynaptic neurons in DMH resulted in the deficiency of autonomic thermoregulation, whereas ablations in PVH and ARC led to alterations of food intake (Fig. 4 and Supplementary Fig. 7). The apMPOA has dense fiber projections to the downstream PVH and ARC, suggesting an anatomical arrangement for temperature-dependent feeding behavior. Among the downstream targets of apMPOA, the PVH is enriched with anorexigenic neurons, such as proopiomelanocortin (POMC) neurons which release anorexigenic α-melanocyte-stimulating hormone (α-MSH)[45,56]. As for the ARC, a key satiety center, is enriched with anorexigenic POMC neurons and feed-promoting NPY/AgRP neurons[37,57,58]. The apMPOA→PVH and apMPOA→ARC pathways, which exhibit warm and cold thermosensitive properties, might play mutually antagonistic roles in regulating feeding behavior. In two recent studies, Deem, et al.[5] and Yang, et al.[59]. found that AgRP neurons activation occurred rapidly upon acute cold exposure and promote food intake, whereas silencing of AgRP neurons selectively blocked the increase of food intake to cold exposure. While Zimmer, et al.[60]. found that thermal stimulus in neonates modulated the functional properties of AgRP neurons in the ARC, with warm temperature blunting the activations of AgRP neurons.

In this study, we further revealed that the neighboring but distinct apMPOA neurons received specific upstream inputs from LPB. The PVH-projecting apMPOA neurons receive transsynaptic inputs from LPBd, whereas the ARC-projecting apMPOA neurons receive anatomical inputs from the LPBe. The LPB → MPOA pathway plays a critical role in transmitting external thermal signals for autonomic thermoregulation[40,41,61–63]. Glutamatergic LPBd neurons transmit warm signals to activate GABAergic MPOA neurons which further suppress thermogenic nuclei. Whereas, cold signals are transmitted from LPBe neurons to GABAergic neurons in MnPO which further inhibit GABAergic neurons in MPOA. In addition to GABAergic neurons, glutamatergic neurons in the MPOA are also involved in warmth sensation and regulation[8,50]. The activation of glutamatergic neurons in the VMPO that are innervated by glutamatergic terminals from LPB, induces body hypothermia[50]. These pioneering findings delineate that feeding behavior, as a means of behavioral thermoregulation[51], shares the same anatomical transmission pathway LPB → MPOA with the canonical autonomic thermoregulation. This issue awaits further experiments of functional manipulations on LPB-MPOA pathways.

**Molecular mechanisms underlying ambient temperature-regulated feeding behavior.** Genetic evidence obtained from projection-specific RNA sequencing showed that the galanin receptor and apelin receptor could be used to molecularly discriminate the two subsets of apMPOA neurons. Galanin receptor neurons are located in multiple hypothalamic nuclei that mediate feeding behavior, including the LH[64], ARC[65], NTS[66], PVH[67], VMH[68], as well as in MPOA[69,70]. Consistent with our findings of hunger-evoked C-Fos expression, galanin receptor neurons in the MPOA were previously reported to be activated by food deprivation[69]. The presence of dense synaptic connections between galanin neurons and both NPY and POMC neurons in the ARC[71], and orexin and melanin-concentrating hormone (MCH) neurons in the LH[72,73], supports the regulatory function of galanin on feeding behavior. Here, our findings further uncovered the functional significance of galanin in the apMPOA in mediating feeding behavior. Galanin receptor neurons were substantially overlapped with apMPOA neurons that specifically project to orexigenic ARC, but not PVH (Fig. 8m, n and Fig.S13a, b). These results suggest that galanin receptor apMPOA neurons→ARC pathway reflects a neural strategy for replenishing energy in response to a cold environment.

Apelin is abundant in hypothalamic nuclei that regulate feeding behavior, including the MPOA, PVH, DMH, ARC, and VMH[74]. Although it is reported that intracerebroventricular injection of apelin inhibits feeding behavior[75–77], the inhibitory effect of apelin on food intake still remains a matter of debate. The hypothalamic corticotropin releasing factor (CRF) system might participate in the apelinergic system, as CRF receptor antagonist could reverse the inhibitory effect of apelin on food intake[75]. Apelin receptor neurons in the hypothalamus are largely colocalized with CRF secreting neurons[78,79]. In another study, apelin receptor neurons were found to be strongly colocalized with POMC, and can regulate feeding behavior through α-MSH release[74]. Apelin is also expressed in the MPOA[80], however, the specific neuronal function remains unclear. According to our findings, the apelin receptor neurons were largely overlapped with apMPOA neurons that specifically project to anorexigenic neurons in the PVH, but not overlapped with ARC-projecting apMPOA neurons (Fig. 8o, p and Supplementary Fig. 13c, d). This finding suggests that apelin receptor apMPOA neurons→PVH pathway reflects another neural strategy for suppressing energy intake in response to a hot environment.

**Ambient peripheral thermal input regulates feeding behavior**. In this study, we found that apMPOA neurons showed obvious responses to ambient exposure but not (or weak) to local exposure (Fig. 6d, e, i–j). The way by which central hypothalamus responds to peripheral local or environmental thermal inputs has yet to be fully elucidated. Diverse central responses induced by local body parts and ambient whole body exposure might be attributed to variations in the density, thresholds, cell types, and afferent pathways of cutaneous thermoreceptors[81,82]. Thermosensation and regulation works independently through various scattered thermal effectors related to diverse thermoregulation variables, but not be completed by a central integrator[81,83]. In terms of the central system, multiple neural circuits exist in the brain, where each thermoregulatory effector system is driven by its own peripheral thermal sensors. For example, local warming or cooling on hands induces local thermal intensity sensation and vascular activity, rather than BAT thermogenesis, or shivering[84,85]. Particularly, local warming or cooling on face or head differs from other sites (hand, leg, etc.), producing changes in higher cognitive or emotional aspects[86,87]. Whereas, ambient thermal exposure results in diverse thermoregulation variables, such as vasodilation, locomotion, energy expenditure, BAT thermogenesis, shivering[50,55,88,89]. In the current study, the coordinating feeding behavior innervated by apMPOA acts as a means of behavioral thermoregulation to maintain energy homeostasis[51], only in coping with ambient challenges that probably disrupt the homeostasis, but not in response to the local thermal stimulus or pain that usually would not affect energy homeostasis (eg., thermal stimulus on the plate in Fig. 6b). This property of apMPOA-innervated pathways suggests an evolutionarily conserved regulatory role of temperature on feeding behavior only in the case of ambient challenges.

In summary, we identified the apMPOA as a previously unknown structure that intertwines ambient temperature and food intake. In receiving distinct synaptic inputs from different parts of LPB, the PVH and ARC-projecting apMPOA neurons that express apelin and galanin receptors orchestrate feeding behavior in coping with ambient thermal challenges. By developing a physiological understanding of the ambient temperature-regulated feeding behavior, this study provides a probable entry point for studying neural mechanisms of feeding disturbance of patients with pathological POA lesions, as well as fever-induced anorexia.

## Methods

**Animals**. Mice were group housed under a 12 h light-dark cycle with ad libitum access to food and water, 50–70% humidity, and 18–22 °C ambient temperature unless otherwise noted. Adult male and female (aged 2–3 months) C57BL/6J, Ai9 (Cre-dependent tdTomato reporter) mice, (Jackson Laboratories, Stock No. 007909), vGluT2-IRES-Cre mice (Jackson Laboratories, Stock No. 016963), and GAD2-IRES-cre mice (Jackson Laboratories, Stock No. 010802) were used in this study. All experimental procedures were approved by the Animal Care and Use Committee of Army Medical University.

**Stereotactic virus injection and optical fiber implantation**. The present study used mice at approximately 2 months of age, with balanced numbers of males and females, and consistent body weight across (22.4 ± 1.2 g) all the groups and experiments. After anesthesia with a mixture of ketamine (55 mg/kg) and xylazine (6.4 mg/kg), the hairs on the scalp of mice were shaved off. Erythromycin ointment was rubbed on the eyes of the mice. Then, the mice were fixed on a stereotaxic device. For the study of retrograde calcium signaling in the apMPOA→ARC pathway in response to a thermal stimulus (Fig. 6), adeno-associated virus AAVretro-Flp (AAV2/retro-DIO-Flp, $5.79 \times 10^{12}$ vg/mL), which expresses recombinant enzymes retrogradely, was injected 60-65 nL unilaterally at a rate of 25 nL/min at the site of the ARC, with the injection coordinates: AP = −1.7 mm, ML = 0.25 mm, DV = 5.65–5.7 mm, with the glass electrode left in place for about 10 min and then withdrawn. Then the glass electrode was replaced and filled with mineral oil. The Cre-dependent GCaMP6s virus (AAV2/8-EF1a-fDIO-GCaMP6s, $5.14 \times 10^{12}$ vg/mL, 100–120 nL), was aspirated and injected unilaterally in the apMPOA at a rate of 25 nL/min, with the injection coordinates: AP = +0.45 mm,

ML = 0.25 mm, DV = 4.9–5.0 mm. In the experiment of calcium signaling of apMPOA to altered dietary states (Fig. 1k), the Cre-dependent GCaMP6s virus (AAV2/8-EF1a-DIO-GCaMP6s, $5.94 \times 10^{12}$ vg/mL) was injected into the apMPOA of vGluT2-ires-cre mice. After virus injection, an optical fiber was placed 200–300 μm over the apMPOA at AP = +0.45 mm, ML = 0.25 mm, DV = 4.5 mm. After the placement of optical fiber, two skull screws were fixed firmly in the skull surface. Then the fiber and screws were completely covered with dental cement. The gripper holding the fiber was removed after the cement had set. The mice were returned to their home cages and housed for 2–3 weeks before testing. For the optogenetic experiment (Figs. 2a and 3a), the Cre-dependent hChR2 virus (AAV2/8-EF1a-DIO-hChR2-EYFP, $1.33 \times 10^{13}$ vg/mL, 100–120 nL) was injected into the bilateral apMPOA of vGluT2-cre mice. Allowing 4 weeks for virus expression, two optical fibers were placed above the apMPOA or its target areas. The detailed coordinates for fiber placements were as follows: apMPOA: AP = +0.45 mm, ML = ±2.0 mm, angle of deflection = 22°, DV = 4.65 mm; PVH: AP = −0.7 mm, ML = ±1.5 mm, angle of deflection = 16°, DV = 4.3 mm; PVT: AP = −1.22 mm, ML = ±1.0 mm, angle of deflection = 19°, DV = 2.7 mm; ARC: AP = −1.7 mm, ML = ±2.3 mm, angle of deflection = 21°, DV = 5.5 mm; LH: AP = −1.3 mm, ML = ±2.0 mm, angle of deflection = 10.5°, DV = 4.5 mm; TN: AP = −1.58 mm, ML = ±1.5 mm, angle of deflection = 5.5°, DV = 5.15 mm; VTA: AP = −3.28 mm, ML = ±1.5 mm, angle of deflection = 14°, DV = 4.0 mm; PAG: AP = −2.7 mm, ML = 0.25 mm, DV = 2.5 mm; DMH: AP = −1.8 mm, ML = ±1.5 mm, angle of deflection = 12°, DV = 4.8 mm; BNST: AP = +0.5 mm, ML = ±2.0 mm, angle of deflection = 21°, DV = 3.1 mm; BA: AP = −2.54 mm, ML = ±2.75 mm, DV = 3.8 mm. In the experiment of genetic ablating postsynaptic neurons of apMPOA (Fig. 4a), a high titer of AAV1-cre-GFP was injected into the bilateral apMPOA ($1.12 \times 10^{13}$ vg/mL, 100–120 nL). The Cre-dependent vector AAV-flex-taCasp3-TEVp was respectively injected into bilateral ARC (at the above injection coordinates, $6.73 \times 10^{12}$ vg/mL, 65nL), PVH (at the above injection coordinates, 65 nL/unilateral), and DMH (AP = −1.8 mm, ML = ±0.32 mm, DV = 5.5 mm, 80 nL/unilateral). In the experiment of chemogenetic inhibition of PVH and ARC neurons, the Cre-dependent vector AAV2/9-DIO-hM4Di-mCherry encoding inhibitory Designer Receptors Exclusively Activated by Designer Drugs (DREADDs, hM4Di) was respectively injected into the bilateral ARC (at the above injection coordinates, $3.6 \times 10^{12}$ vg/mL, 65nL), PVH (at the above injection coordinates, 65nL/unilateral). Notably, high titer of AAV1, as an anterograde transsynaptic tool for tagging postsynaptically targeted neurons, is probably taken up by terminals and is back transported to the upstream nucleus[90]. Therefore, prior to the experiment for Figs. 4 and 5, we had performed control experiments for AAV1 to confirm that there is no reverse projection from apMPOA to PVH and ARC. We injected anterograde AAV-syn-GFP into the PVH or ARC to investigate whether there are terminals in the apMPOA innervated by PVH or ARC (Supplementary Fig. 6a, b, d). The result showed that no or very sparse projections from PVH or ARC to apMPOA (Supplementary Fig. 6c, e). Additionally, we injected retrograde AAVretro-syn-GFP in the apMPOA to investigate whether there were retrogradely labeled neurons in PVH or ARC (Supplementary Fig. 6f–i). The results further demonstrated there were no substantial projection from PVH or ARC to apMPOA.

**Optogenetic stimulation parameters**. The parameters of 473 nm laser for optogenetic activation were set as: 10 ms light pulse at 20 Hz in 1 s ON and 3 s OFF duty cycle with laser power 5 mW at fiber tip. The duration of optogenetic stimulation: 2 h for the feeding behavior test, 60 min for the rectal temperature recording, and 30 min for the infrared thermography. Animals were sacrificed after completion of experiments to confirm specificity of viral expression and fiber placement.

**Temperature exposure test**. To assess the calcium activity of the apMPOA induced by thermal stimulation, two thermal stimulation approaches were designed. One involved the application of a localized temperature stimulus on an aluminum plate via semiconductor cooling/heating. The mice could move around on the aluminum plate, and local temperature stimulation was applied on their feet when the temperature changed. The temperature of warm plate and cold plate were designed as 40 and 10 °C according to previous studies[17,32]. In the other method, mice were subjected to whole-body temperature exposure in thermostatically controlled chambers. In contrast to the thermal plate temperatures, the temperatures of the two chambers were set at 35 and 10 °C to avoid overheating-induced mania or death. The preset temperatures of the thermal plates and chambers could be quickly reached from room temperature 25 °C in about 3 min via semiconductor temperature control module.

**Temperature recording**. Rectal temperature was measured by using a high-sensitivity digital thermometer (GM1365) with a precision of 0.1 °C. We inserted the tip of the thermometer into the mouse anus and recorded the temperature reading while it remained steady. The BAT temperature was dynamically monitored with an infrared imager (FLIR ONE). The infrared imager was turned on for dynamic video recording when the mice were optogenetically stimulated. After the experiment, infrared images at the specified time points were captured.

**Fiber photometry**. In the present study, the calcium activity in the apMPOA induced by fasting and ambient temperature exposure was recorded using a fiber photometry system (C1410488, Inper, Hangzhou). The ceramic ferrules placed into the mouse brain were connected to 410 and 488 nm laser. The activity-dependent GCaMP6s fluorescence signal (488 nm) and activity-independent fluorescence (410 nm) were all captured by a CMOS camera, providing internal control for movement and bleaching artifacts. For $\Delta F/F$ calculation, baseline $F_0$ was defined as the average $F$ during 4 min before feeding behavior was initiated in the fasting-induced calcium activity recording experiment, and $F_0$ was defined as the average $F$ during 5 min before the ambient temperature was changed in the temperature-induced calcium activity recording experiment.

**Feeding behavior test**. Wild type mice were deprived of food, but not water, overnight (8:00 pm–8:00 am +1). In the fasting-induced C-Fos experiment, the re-feeding group was presented with standard chow for 2 h after overnight fasting, while the fasting group was not presented with chow. The no-fasted group was not deprived of food. 2 h later, the three groups were all sacrificed and perfused. In the optogenetic, chemogenetic, and ablation experiments, the mice were deprived of food, but not water, for 12 h (8:00 am–8:00 pm). Then they were presented with the standard mouse chow in the feeders. Two-hour food intake was recorded. Regarding food intake during ambient temperature exposure, both two-hour and twelve-hour food intake were all recorded after fasting from 8:00 am to 8:00 pm. In the fiber photometry experiment of calcium signals in response to dietary state, the mice were fasted for 24 h in order to ensure that they would eat during the recording process. Of note, mice that experience cold exposures and are deprived of food for multiple times would likely develop mild torpor or hibernation[18,19,25], and have a high risk of sickness, exhibiting significant decreased food intake, loco-motion, and rectal temperature during cold exposure. The data of mice with mild torpor or sickness should be discarded.

To avoid leaving behind chow crumbs and mixing with mouse droppings and urine, a mouse feeder was designed. We drilled four semicircular holes in two rectangular copper blocks (designed according to the feed diameter, 9 mm). The size of the copper block was 8 cm × 1 cm × 2.5 cm. The chow was placed into the semicircular holes, and the two copper blocks are then tightened with bolts. The length of chow protruding from the copper did not exceed 6 mm. In order to collect chow crumbs, a narrow metal trough under the copper block was designed. The size of the metal trough was 8 cm × 1.5 cm × 0.5 cm. After feeding behavior test, the remaining intact chow and crumbs are weighed together and the difference in weight from the pre-feeding weight was calculated as the food intake. The food intake was obtained by averaging the amount of food consumed by the mice across three times.

**Immunohistochemistry**. The mice were anesthetized by the intraperitoneal injection of 25% urethane (1.5 mg/kg). Then, they were perfused with 0.1 M PBS, followed by 4% paraformaldehyde (PFA). After the mouse was well fixed, the mouse brain tissue was removed and placed in 4% PFA overnight for further fixation. The brain tissue was immersed in 10% sucrose solution (4% PFA) on the following day and dehydrated for 2 days. For complete dehydration we immersed the tissue in 20 and 30% sucrose solution for gradient dehydration. The brains were frozen and sectioned into 30 μm slices. In the C-Fos staining experiment, after washing in 0.1 M PBS with 0.3% Triton-X 100 for 30 min and blocking with 10% goat serum for 1 h at 37 °C, the sections were incubated with the primary antibody, anti-cFos (1:1000, rabbit, ab190289, Abcam), at 37 °C for 30 min and then at 4 °C overnight. After washing in 0.1 M PBS for 30 min, the sections were incubated with a fluorescent secondary antibody (goat anti-rabbit 488, 1:500, ab150077, Abcam, or goat anti-rabbit 568, 1:500, ab175471, Abcam) at 37 °C for 1 h. Finally, all the sections were stained with DAPI at room temperature for 10 min and washed in 0.1 M PBS for 30 min.

**Anterograde-Tracing**. A Cre-dependent rAAV expressing the hChR2-eYFP protein (AAV8-EF1α-DIO-hChR2-eYFP) was bilaterally injected into the apMPOA ($1.33 \times 10^{13}$ vg/mL, 100–120 nL). After injection, the mice were returned to their home cages and kept for 4 weeks. The animals were sacrificed after night fasting (8:00 pm–8:00 am +1) and processed for C-Fos immunohistochemical staining.

**Retrograde labeling**. To retrogradely label the neurons in the apMPOA from the PVH and ARC, we injected the AAVretro-cre virus into the PVH and ARC of Ai9 respectively ($5.24 \times 10^{12}$ vg/mL, 65 nL). The Cre recombinase was expressed retrogradely from the fiber terminals to the cell bodies in the apMPOA and combined with the fluorescent protein tdTomato. The labeled cell bodies were further quantified for the colocalizations with the C-Fos induced by thermal stimulations.

To differentiate the neurons in the apMPOA that project to the PVH and ARC, we likewise injected CTB-555 (Invitrogen, C-34775, 1.0 mg/mL, 65 nL) and CTB-488 (Invitrogen, C-34775, 1.0 mg/mL, 65 nL) into the PVH and ARC respectively in wild-type mice. The two CTBs were retrogradely expressed from the terminals to cell bodies in the apMPOA. The colocalization of neurons projecting to the PVH and ARC was quantified.

To explore the upstream inputs of PVH and ARC-projecting apMPOA neurons, we performed monosynaptic retrograde labeling using modified rabies

virus. The helper viruses AAVretro-DIO-TVA-EGFP and AAVretro-DIO-oRVG were mixed in equal volumes and 70 nL of the mixture was then unilaterally injected into the ARC and PVH of vGluT2-cre mice respectively ($5.24 \times 10^{12}$ vg/mL). After expression for three weeks, the rabies virus RV-ENVA-ΔG-dsRed ($2.5 \times 10^8$ IFU/mL, 100 nL) was injected into the apMPOA. Then, the mice were returned to their home cages. After one week of viral replication, the mice were sacrificed and their brains were prepared for further imaging.

**Projection-specific RNA sequencing**. To molecularly differentiate the neurons in the apMPOA projecting to the ARC and PVH respectively, we performed transcriptome sequencing using the SmartSeq2 protocol[44]. The neurons in the apMPOA were specifically labeled by the injection of Cre-dependent AAVretro-DIO-EGFP into the ARC and PVH of vGluT2-cre mice ($3.8 \times 10^{12}$ vg/mL). In order to accurately collect the GFP-labeled neurons in apMPOA, but not neurons in neighboring nuclei, we collected the cells in the acute brain sections under a microscope with a glass pipette, rather than fluorescence activated cell sorting[90]. After allowing three weeks for the expression of EGFP, the mice were sacrificed and their brains were quickly removed and submerged into ice-cold ACSF. Then the brains were cut into 300 μm-thick coronal slices containing the apMPOA, PVH, or ARC. The slices containing apMPOA were incubated in N-methyl-D-glucamine (NMDG)-incubation fluid (110 NMDG, 110HCl, 2.5 KCl, 1.2 NaH$_2$PO$_4$, 25 Glu-cose, 10 MgSO$_4$, 0.5 CaCl$_2$ and 25 NaHCO$_3$ in mM) for 15 min at 32 °C. Then the slices were transferred to oxygen-saturated ACSF and incubated for 1 h at room temperature. After incubation, the slices were transferred to the recording chamber of an electrophysiological recording platform. Oxygen-saturated ACSF was continuously perfused.

To collect fluorescently labeled neurons, glass pipettes of 2–4 MΩ were used to suck the neurons under a microscope. The pipettes were filled with sterile PBS and a slight positive pressure was given before attaching the cell. The pipette tip approached the cell from the side of the cell, rather than from the top. Once the tip touched the cell, we quickly removed the positive pressure and gave a light negative pressure until the cell had been visibly sucked into the tip. Then, we locked in the negative pressure and raised the pipette away from the brain slice. We broke the tip of the pipette into a PCR tube containing 4 μl RNase inhibitor in lysis buffer. The PCR tube was store on ice until 15 cells were collected for one mouse sample. Then the tube was transferred to −80 °C ultra-low refrigerator. A total of five biological repeats were collected for the PVH and ARC groups respectively.

The SmartSeq2 protocol was used to perform RNA sequencing. The samples were incubated at 72 °C, and immediately returned to the ice. Then the samples were reversely transcribed to cDNA based on the polyA tail and amplified the full-length cDNA by PCR. RNA degradation and contamination were monitored by 1% agarose electrophoresis. The library construction was prepared according to the Tagmentation-based library protocol. The PCR products were purified and selected with the Agencourt AMPure XP-Medium kit. DNA was quantified with an Agilent Technologies 2100 bioanalyzer. Then the library was amplified to generate DNA nanoball (DNB), containing more than 300 copies of a single molecule. The DNBs were loaded into the patterned nanoarray and single end 50 bases reads were generated in the way of sequenced by combinatorial Probe-Anchor Synthesis (cPAS). In this project, a total of 10 samples were measured, the libraries were multiplexed and sequenced on an DNBSEQ platform, and 100 bp paired-end reads were generated, and each sample produced an average of 6.83 G data.

HISAT2 (v2.0.4) was used to map the reads to the mouse genome for transcript abundance quantification (mm10, build name GRCm38). Transcripts per kilobase of exon model per million mapped reads (TPM) or fragments/kbp of transcript/ million mapped reads (FPKM) values of each gene were calculated based on the length of the gene and the number of reads mapped to the gene. The correlational analysis, enrichment analysis and clustering analysis of DEGs were carried out on the Dr. Tom system (an interactive report system developed by BGI). DESeq2 was conducted to identify DEGs between the ARC and PVH groups, and significant DEGs were selected as fold change ≥ 2 and q ≤ 0.05. The resulted DEGs were annotated using the database Kyoto Encyclopedia of Genes and Genomes (KEGG) to obtain further information. According to the KEGG_pathway annotation classification, we used the Phyper function in R package to perform the enrichment analysis (https://stat.ethz.ch/R-manual/R-devel/library/stats/html/Hypergeometric. html), and calculated the $P$ value. The Q value was obtained by the resulting $P$-values that were adjusted for controlling the false discovery rate using the Benjamini and Hochberg's approach (https://bioconductor.org/packages/release/bioc/html/qvalue.html). The function of Q value ≤ 0.05 is regarded as a significant enrichment. The version of KEGG pathway analysis was 93.0 obtained from KEGG database (93.0/genes/ko/ko_pathway.list).

**Multiplex fluorescence in situ hybridization**. We performed in situ hybridization (ISH) using the RNAscope kit (Advanced Cell Diagnostics). The FISH was performed with probes targeting Aplnr (GenBank: NM_011784.3, RNAscope®Probe-Mm-Aplnr), GalR1 (GenBank: NM_008082.2, RNAscope®Probe-Mm-Galr1-C2) and Fos (GenBank: NM_010234.2, RNAscope®Probe-Mm-Fos-C3) mRNA. We labeled the projection specific glutamatergic apMPOA neurons by injecting Cre-dependent AAVretro vectors expressing mCherry into the PVH or ARC (not in the same mouse) of vGluT2-cre mice. After expression for 3 weeks, the mice were overnight fasted and sacrificed. Mice were perfused with the brain tissue placed in

4% PFA, and gradually dehydrated by sucrose as described above. Brains were cut coronally at 20 μm, and stored at −80 °C before processing. RNAscope experiments were performed as indicated in user manual. Sections were washed with ethanol, followed by tissue pretreatment, probe hybridization, and signal amplification. Aplnr, GalR1, and Fos probes were diluted 1:50 in probe diluent.

**Pharmacological experiments**. In pharmacological experiments of injecting agonists and antagonists of candidate genes, wild-type mice were implanted with cannulas above the bilateral apMPOA (AP = +0.45 mm, ML = ±2.0 mm, angle of deflection = 22°, DV = 4.7 mm) which were secured to the skull with screws and dental cement. Then the mice were returned to home cages and housed for 1 week for recovery. Before behavioral testing, the mice were habituated to handling, injection, and food feeder for 2 h every day. Drugs were prepared fresh from frozen aliquots before each experiment and were then microinjected (150 nL/unilateral) into the apMPOA with an internal cannula connected to a microliter syringe pump. Fifteen min after bilateral injections, the mice were returned to their home cages and feeding behavior tests were performed for 2 h. The drugs were prepared fresh on the day of the experiment by dilution in vehicle solution (PBS supplemented with 0.1% DMSO, 5% PEG400, and 5% Tween-80), including the integrin inhibitor SB273005 (Selleck, 150 nL/unilateral, 1 μM), the enolase inhibitor AP-III-a4 (ENOblock, Selleck, 150 nL/unilateral, 1 μM), the IRAK4 inhibitor PF-06650833 (Selleck, 150 nL/unilateral, 1 μM), the antagonist of apelin receptor ML221 (Selleck, 150 nL/unilateral, 1 μM), the serotonin reuptake inhibitor Fluvoxamine maleate (MK-264, Selleck, 150 nL/unilateral, 1 μM), the antagonist of dopamine D3 receptor BP897 (Adooq Bioscience, 150 nL/unilateral, 1 μM), the antagonist of NPY Y1 receptor BIBO 3304 trifluoroacetate (Tocris, 150 nL/unilateral, 1 μM) and antagonist of galanin receptor M40 (Tocris, 150nL/unilateral, 1 μM). Bioactive neuropeptides were prepared fresh by dilution in sterile PBS, including apelin receptor agonist apelin-13 (Tocris, 150 nL/unilateral, 1 μM), neuropeptide Y (Tocris, 150 nL/unilateral, 1 μM) and galanin receptor agonist galanin(1–29) (Tocris, 150 nL/unilateral, 1 μM).

**Statistics**. Animals were randomly assigned to control and treatment groups. For animals with multiple treatments, the sequence of treatments was randomized. Data are represented as mean ± standard error of the mean (SEM). SPSS 18.0 and Prism 8 software (GraphPad) were used for statistical analysis and graphing. T test, paired t test, one-way ANOVA and two-way repeated-measure ANOVA and post hoc least significant difference (LSD) multiple comparisons were used to test significance between samples. Significance was defined at a level of $P < 0.05$.

**Reporting summary**. Further information on research design is available in the Nature Research Reporting Summary linked to this article.

## Data availability
The RNA-seq data have been deposited in the database of the NCBI Sequence Read Archive (https://www.ncbi.nlm.nih.gov/sra/) under the accession number PRJNA820979. All raw data generated from RNA-sequencing are freely available from the NCBI. The mouse genome (mm10, build name GRCm38) can be accessible through NCBI Assembly (https://www.ncbi.nlm.nih.gov/assembly/). RefSeq assembly accession: GCF_000001635.26. The version of KEGG pathway database is 93.0, and can be accessible through KEGG databases (https://www.genome.jp/kegg/pathway.html). Source data are provided with this paper.

## Code availability
Customized Matlab codes were used to analyze the fiber photometry signal and can be accessible (https://github.com/qswenwen/Fiber-Photometry-preprocessing).

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

## Acknowledgements

This work was supported by the National Natural Science Foundation of China to Yi Zhou (Grant No. 31970932, 32171001) and Ying Xiong (Grant No. 31871075, 32071015, 31921003). We thank Professor Li I. Zhang, Professor Huizhong Whit Tao, Dr. Guangwei Zhang, and Dr. Can Tao at University of Southern California for advice on viral tools, Dr. Zhen Zhao at University of Southern California and Dr. Xinying Guo at Guangzhou Medical University for advice on RNA sequencing experiments, Dr. Min Xu at Institute of Neuroscience, Chinese Academy of Sciences and Dr. Xiaowei Chen at Army Medical University for helpful discussions.

## Author contributions

Y.Z., Y.X., and S.Q. conceived the study. S.Q., S.Y., and R.P. performed most behavioral experiments. S.Y. and X.H contributed to the identification of animal genotypes. J.Z. and

K.L. designed the thermal plate and chamber. S.Q., Z.S., Z.W., and T.L. contributed to recording experiments. Y.Z. (Yanjie Zhang) and S.Q. contributed to data analysis. P.C. and S.Q. contributed to anatomical experiments. Y.Z. and Y.X. supervised the project. Y.Z., Y.X., and S.Q. wrote the manuscript.

## Competing interests

The authors declare no competing interests.

## Additional information

**Peer review information** *Nature Communications* thanks Tune Pers, Roger Adan and Other anonymous reviewer(s) to the peer review of this work. Peer review reports are available.

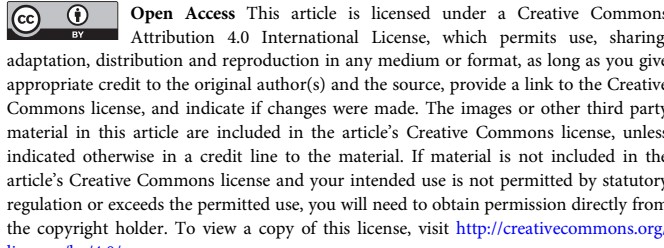

