## [Peer Review File · Nature Communications]

REVIEWER COMMENTS

Reviewer #1 (Remarks to the Author):

The relationship between temperature and feeding has been known for a long time. Feeding increases when ambient temperature is low because energy demands are high, whereas feeding decreases when ambient temperature is high because energy demands are low. In “A temperature-regulated circuit for feeding behavior” Qian and colleagues identify two populations of glutamatergic neurons in the apMPOA that are sensitive to ambient temperature and induce changes in feeding. Collectively, the findings suggest that one population differentially expresses apelin receptor, projects to PVH, is responsive to warmth, and decreases feeding when activated. The second population differentially expresses galanin receptor, projects to ARC, is responsive to cold, and increases feeding when activated. Together these findings reveal two circuits emanating from apMPOA that could explain the effects of temperature on food intake.

The experiments are thorough and presented in a very logical order. In particular, the viral strategy using RV to specifically label the anatomical inputs to apMPOA neurons that projecting to PVH and ARC is clever and informative. The optogenetic activation experiments are also nicely controlled, with important controls that receive light without Chr2. However, there is insufficient evidence to substantiate the claim that “Cutting off both pathways can eliminate the temperature-dependence of feeding.” This is the one major limitation of the manuscript and central to the point of the paper. Without this robust result, it is formally possible that these neurons modulate another type of feeding, not temperature-induced food intake. Although the temperature sensitivity aligns with the direction of the effect of feeding in both populations, as presented the data do not show that the effects of temperature on these neurons modulate their effects on feeding.

Major point:

The authors claim that the pathways identified here are necessary for the effects of ambient temperature on feeding. This is not substantiated by statistics. To address this major limitation, it is essential that the authors present ANOVA statistics, specifically a significant effect of the interaction between temperature and treatment group. Is there a significant interaction? If so, is the LT>NT>HT pattern of food intake significant in the controls but not ablated mice? More generally, the effect of LT on feeding seems to be absent in a lot of the experiments. Why? This is a problem for manipulations, like the caspase ablations, in which the predicted outcome is a modulation of the effect of temperature.

Another consideration related to the link between temperature and the roles of these neurons in feeding is their responsiveness to metabolic state. Lines 52-54 state that “In the instances of extremely hot environments, the lower thermal demand and higher dissipation load preferentially suppress food intake, overriding the dietary state.” The effects of fasting or refeeding on cFOS are only measured at room temperature. The effects of ambient temperature on calcium activity are only measured at one ambient temperature. Wouldn't the quoted statement predict that temperature manipulations would alter the effects of fasting on cFOS? This might be helpful if the effects of caspase ablation are unaltered by temperature.

Minor points:

It is unclear what is meant by: “A similar feeding pattern induced by pharmacological apelin application with optogenetic/chemogenetic manipulations on apMPOA→PVH pathway indicated that the *Aplnr* might be a potential marker for the PVH-projecting apMPOA neurons.” Please rephrase.

All Figures should have legends in the panels, not just in the text.

More methodological clarity is required, such as experimental details, injection volumes, mouse strains, sexes, body weights. For example, body mass can influence all of the variables measured, and so it should be clear that this and other characteristics of the mice are consistent across groups and across experiments.

Reviewer #2 (Remarks to the Author):

Summary:

The overarching aim of the work by Qian et al. was to identify neurocircuits driving temperature-regulated feeding behavior during high and low ambient temperature levels. Using c-fos stainings, glutamatergic and GABAergic reporter mice and fiber photometry they identified two temperature-responsive feeding-regulatory glutamatergic neuronal populations in the anterior periventricular portion of the medial preoptic area (apMPOA). Qian et al. validate the thermoregulatory property of the two apMPOA glutamatergic neurons using optogenetic activation. They found that activation of these two neuronal populations reduced food intake. Using AAV-based anterograde tracing they then show that apMPOA vGlut2-positive cell populations project to multiple brain areas, several of which were activated upon food restriction, including the arcuate nucleus (ARC) and paraventricular nucleus of the

hypothalamus (PVH). Activation of the apMPOA→ARC population promoted feeding in the presence of cold temperatures (increases food intake by >60%), whereas the apMPOA→PVH-projecting cell population suppressed food intake in the presence of hot temperatures. The further characterize the two focal apMPOA cell populations, Qian et al. utilized projection-specific RNA sequencing of manually-collected fluorescently-labelled ARC and PVH-projecting apMPOA cells. The two projecting populations exhibited discrete transcriptional programs. They microinjected inhibitors and agonists of apelin and galanin into the apMPOA and were able to show that these two neuropeptides are likely to act through the bespoke two apMPOA cell populations to regulate temperature-dependent food intake.

Comments:

The authors have performed an extensive set of experiments and their findings are presented in a concise and well-written manner. My comments are focused on the single-cell aspects of the manuscript and I will have to leave it to the other reviewers to assess the remaining parts of the manuscript.

The selection of the galanin and apelin receptors based on the expression data results seems somewhat arbitrary. Were other neuropeptides tested for an effect on food intake?

It would be helpful to know which marker genes differentiate the two apMPOA glutamatergic cell populations from other apMPOA glutamatergic cell populations. It probably would be helpful if the authors could try to map the transcriptional signatures of their two populations onto the Moffit et al single-cell atlas of the preoptic nucleus of the hypothalamus (<https://www.ncbi.nlm.nih.gov/pmc/articles/PMC6482113/>) and show which of the Moffit cell populations their two populations correspond to. This step might also allow them to identify where within the apMPOA their cells are located (Moffit et al provide a spatial atlas along with the single-cell RNA-seq atlas).

In general, a bit more information on these cell populations' transcriptome would be helpful. How many unique genes were identified for each of the two cell populations? How many gene sets were tested for enrichment? Were other databases than GO and KEGG tested? How were they adjusted for multiple testing?

The same applies for the RNA sequencing part of the Methods. What was the sequencing depth? Is the "Dr. Tom system" part of the BGI software suite? Which version of KEGG and GO were used? What was the gene background used in the gene set enrichment analysis?

The names of all differentially expressed genes should be made available along with their test statistics and normalized expression values.

All gene set enrichment results should be made available as source data.

All expression data should be made available.

All source code used to process and analyze the data should be made available to allow others to reproduce their results and to learn from their setup.

I hope that you will find these comments useful. Great work!

Kind regards

Tune H. Pers

Reviewer #3 (Remarks to the Author):

This is an extensive paper investigating the role of MPOA neurons in thermosensitive responses and feeding. The authors start with showing activation of these neurons upon manipulation of ambient temperature and then went on to show that different projection neurons serve different functions. My enthusiasm for this paper was tempered by the fact that I miss several essential controls that support that their targeting approaches are specific and indeed doing what they believe they do.

Major comments:

Can you exclude that axonal stimulation leads to backpropagation of activation and spread to other projection areas (thus are there multiple innervation targets from a single neuron?)

The data in figure 4 require more controls. The authors claim AAVcreGFP jumps anterogradely but there are several alternative explanations. It could also be that this virus is taken up by terminals and is

backtransported to the nucleus of cells in the MPO target areas (thus neurons projecting to MPO may have been targeted). There may be very limited leaky expression in absence of cre.

Also for the Rabies experiments controls are needed without TVA injected in Glut-cre mice. Also all virus should be injected in wt mice to exclude leaky expression

I recommend RNASCOPE experiments to verify VGlut, apelin receptor, galanin receptor in cells that get fos activated by fasting combined with projection specific markers to verify the identity of the projection specific glutamatergic neurons

For all experiments details on success of targeting nuclei with viruses are missing. How many mice were excluded from the experiments due to missed injections? Were there unilateral hit mice when the purpose was bilateral hit? What was the total amount of mice used for these experiments?

Other comments:

The grammar of the manuscript needs to be improved by a native English speaking scientist. In some cases poor language results in wrong statements such as in line 408 "...which induce proopiomelanocortin (POMC) neurons..." Which does not make sense. Which are activated by POMC neurons is probably meant.

How was proximity between cFos and ChR axons determined

To which cells in arc do Glu neurons project to, likely AgRP as FI was increased?

For the SmartSeq experiments what is a biological replicate? Is it 5 mice?

Why was FACS sorting not done in order to perform single cell level analyses?

The discussion should be written more condense and focus on the most important findings. There is also some overinterpretation such as: the LPB-MPOA-ARC pathway was anatomically mapped, but the authors did not demonstrate that LPB provide temperature information to the neurons engaged in feeding responses. The only show connectivity.

Line 435: “ These findings revealed an unrecognized neural circuits of temperature-regulated feeding behavior orchestrated by LPB→apMPOA→PVH/ARC pathways” is such overinterpretation.

Experimental details are missing such as injected volume of Cre-dependent hChR2 virus (AAV2/8-EF1a-DIO-hChR2-EYFP)

Since 40C is warm but not hot, I propose to replace hot plate with warm plate (as hot plate is also used to induce pain).

Concentrations of CTB-555 and CTB-488 are not mentioned

For the optogenetic stimulation in fig 2 what were the parameters (light intensity, frequency etc etc)

In fig 3 were the same mice used at different ambient temperatures? What were the conditions for optostim?

Fig 6c suggest that a single neuron innervates PVN and Arc, how does this impact on the results of optostim of target regions?

The title of Fig. S2 is misleading: C-Fos expressions in multiple thermosensitive preoptic and 984 hypothalamic nuclei in a hunger state; the authors did not determine in this study whether they were thermosensitive

Response to Reviewer 1:

Comment: *The relationship between temperature and feeding has been known for a long time. Feeding increases when ambient temperature is low because energy demands are high, whereas feeding decreases when ambient temperature is high because energy demands are low. In “A temperature-regulated circuit for feeding behavior” Qian and colleagues identify two populations of glutamatergic neurons in the apMPOA that are sensitive to ambient temperature and induce changes in feeding. Collectively, the findings suggest that one population differentially expresses apelin receptor, projects to PVH, is responsive to warmth, and decreases feeding when activated. The second population differentially expresses galanin receptor, projects to ARC, is responsive to cold, and increases feeding when activated. Together these findings reveal two circuits emanating from apMPOA that could explain the effects of temperature on food intake. The experiments are thorough and presented in a very logical order. In particular, the viral strategy using RV to specifically label the anatomical inputs to apMPOA neurons that projecting to PVH and ARC is clever and informative. The optogenetic activation experiments are also nicely controlled, with important controls that receive light without ChR2. However, there is insufficient evidence to substantiate the claim that “Cutting off both pathways can eliminate the temperature-dependence of feeding.” This is the one major limitation of the manuscript and central to the point of the paper. Without this robust result, it is formally possible that these neurons modulate another type of feeding, not temperature-induced food intake. Although the temperature sensitivity aligns with the direction of the effect of feeding in both populations, as presented the data do not show that the effects of temperature on these neurons modulate their effects on feeding.*

Answer: Thank you for your concerns regarding the temperature dependence of feeding behavior. These are insightful comments, and we couldn't agree more. In our original manuscript, we had paid too much attention to the regulatory effect of apMPOA-related neural circuit on feeding behavior and the thermal characteristics of apMPOA-related neural circuit, but had not systematically analyzed the changes of the regulatory effects of apMPOA-related neural circuit under different ambient temperatures, especially the interaction effects between temperature and treatment as you raised. In response to your concerns in **Major point 1** and **point 2**, we carried out additional experiments, data updates, statistical analysis and necessary explanations to support the temperature dependence of feeding as follows:

Firstly, in response to incomplete statistical analysis raised in the **Major point 1**, we further added statistical descriptions to clarify the interaction effects (temperature × treatment) in the two-way repeated-measure ANOVA. The results showed significant interaction effects between temperature and treatment in the experiments of optogenetic activating, caspase ablating or chemogenetic inhibiting of apMPOA-recipient PVH neurons, but not in the apMPOA-recipient ARC neurons. These statistical results support the view that manipulating the apMPOA→PVH would disturb the normal temperature dependence of feeding behavior. Although we did not find that caspase ablating or chemogenetic inhibiting apMPOA→ARC pathway could

eliminate the temperature dependence of feeding, in a very recent study, Yang et al. [1] found that specifically silencing the MPOA-projecting AgRP neurons in the ARC selectively blocked the increased food intake to cold exposure. We speculate that temperature dependence to cold exposure needs specific neuronal apMPOA→ARC pathway. This awaits further identification in future studies. In the revised version, we have revised the related descriptions in the abstract, results, discussion and corresponding figure legends. The detailed response can also be seen in the *Answer to Major point 1*.

Secondly, regarding to the absence of LT > NT pattern of food intake in several experiments raised in the *Major point 1*, we carefully checked the relevant raw data of food intake, rectal temperature and video recordings. We found several mice showed singular values of food intake during low temperature during the second and third feeding tests. One possible explanation for the singular values might be torpor induced by multiple cold exposures and food deprivations. Animals that experienced cold exposures or food deprivation would likely develop a mild form of daily torpor or hibernation. Animals in torpor state are characterized by decreased metabolic rate, body temperature, locomotion and food intake [2-4]. We checked the rectal temperature of the mice with reduced food intake during the second and third feeding tests, and found that these mice showed decreased rectal temperature compared with the other mice during the second and third cold exposures. Therefore, we speculated that the food intake with absences of LT>NT might be singular values caused by mild torpor. Another possible explanation might be that they were sick at the second and third feeding test. Multiple cold exposures and food deprivations increased the risk of sickness which might decrease their appetite. In the *Answer to Major point 1*, we showed the singular values in the raw feeding data. In the revised version, we discarded data from the mice in torpor state or sickness as singular values, and performed additional experiments to improve the data robustness. In the revised manuscript, the effect of LT on food intake was significant after eliminating the influence of singular values. More details can be found in the *Answer to Major point 1*.

Thirdly, in response to the concern whether temperature manipulations would alter the effects of fasting on C-Fos raised in the *Major point 2*, we performed additional experiments to examine fasting and refeeding induced C-Fos at different temperatures. The results showed that C-Fos was substantially expressed at ambient low and high temperatures (revised Figure S3). Similar results can be also found in the supplementary results of the study by Zheng-Dong Zhao [5]. Possible explanation for this is the large distribution of thermosensitive neurons in apMPOA. It appears that the effect of fasting and refeeding on C-Fos expression is masked by the effect of ambient exposures, and it is difficult to distinguish between the effect of fasting/refeeding or thermal exposure on C-Fos expression. Therefore, in our original manuscript, we showed the effect of fasting and refeeding on C-Fos at normal temperature, but not high or low temperature. In the revised manuscript, we have supplemented the C-Fos expressions at different ambient temperatures.

More detailed information can be seen in the *Answers to Major point 1 and 2*. We

hope that these explanations, data updates, statistical revisions, and extra experiments for the temperature-dependence of feeding behavior could address your concerns.

Major points:

Question 1: *The authors claim that the pathways identified here are necessary for the effects of ambient temperature on feeding. This is not substantiated by statistics. To address this major limitation, it is essential that the authors present ANOVA statistics, specifically a significant effect of the interaction between temperature and treatment group. Is there a significant interaction? If so, is the LT>NT>HT pattern of food intake significant in the controls but not ablated mice? More generally, the effect of LT on feeding seems to be absent in a lot of the experiments. Why? This is a problem for manipulations, like the caspase ablations, in which the predicted outcome is a modulation of the effect of temperature.*

Answer: We apologize for the incomplete ANOVA statistics in our original descriptions. We agree with the reviewer that it would be helpful to show the effects of ambient temperatures on both neuronal populations in modulating feeding. We performed two-way repeated-measure ANOVA on the food intake data from the optogenetic experiment (Figure 3d-f), caspase ablations experiment (Figure 4c, 4e), and chemogenetic experiment (Figure 4h, 4i, 4k, 4l). We found significant interaction effects between temperature and treatment in the experiments of optogenetic activation, caspase ablation and chemogenetic inhibition of apMPOA-recipient PVH neurons. Further within-group tests showed a significant simple effect of temperature in control group, but not in the taCasp3 or chemogenetic inhibiting group. And further post hoc pairwise comparisons showed no significant differences among three thermal conditions. This meant that the LT>NT>HT pattern of food intake was not significant in the mice with ablating or inhibiting apMPOA-recipient PVH neurons as the controls. These statistical results support the view that manipulating the apMPOA→PVH would disturb the normal temperature dependence of feeding behavior. However, in the mice with optogenetic activation, caspase ablation or chemogenetic inhibition of apMPOA-recipient ARC neurons, there were no significant interaction effects between temperature and treatment. Although we did not find that caspase ablating or chemogenetic inhibiting apMPOA→ARC pathway could eliminate the temperature dependence in cold exposure of feeding, in a very recent study, Yang et al. [1] found that specifically silencing the MPOA-projecting AgRP neurons in the ARC selectively blocked the increased food intake to cold exposure. We speculate that temperature dependence to cold exposure needs specific neuronal apMPOA→ARC pathway. This awaits further identification in future studies. In summary, in the revised manuscript, we added detailed descriptions about the interaction and main effects of two-way repeated-measure ANOVA statistics in the legends of Figure 2, 4, 7, S1, S4, S5. We also revised the related descriptions in the abstract, result, and discussion.

Regarding to the absence of LT > NT pattern of food intake in several experiments, we statistically compared the food intake between different thermal conditions, and found that the LT > NT pattern of food intake was absent in the control groups in

Figure 3f, Figure 4c, 4h, 4k (original submission). We checked the raw data to find the cause of the absence. In our former version, the food intake was obtained by averaging the amount of food consumed by the mice across three times. We found that the promoting effects of low temperature on food intake were attenuated at second and third times in several mice. Taking Figure 4c for example, the raw food intake data across three times feeding tests are depicted as the following figure. Two of the six mice showed stable food intake across three times at neutral temperature, but showed singular food intake at the second and third times at low temperature. Our video recordings taken during the behavioral experiment further showed that the two mice were inactive and crouching in the corner for most of time during the second and third tests.

Figure a, the raw food intake at neutral temperature during three times feeding tests for the control group in the Figure 4c (gray boxes). **b**, the raw food intake at low temperature during three times feeding tests for the control group in the Figure 4c. Two mice showed decreased food intake during the second and third tests that were depicted in dashed line. **c**, the raw rectal temperature at low temperature during three times feeding tests. The corresponding two mice showed lower rectal temperature during second and third tests.

We think that the possible explanation for the singular food intake at second and third tests for several mice might be torpor or sickness induced by multiple cold exposures and food deprivations. Animals that experience cold exposures and food deprivation for multiple times are likely to develop mild torpor or hibernation. Animals in torpor state are characterized by decreased metabolic rate, body temperature, locomotion, sensory perception, and food intake [2-4]. We checked the rectal temperature of the mice with reduced food intake during the second and third cold exposures, and found that these mice showed decreased rectal temperature compared with the other mice. Therefore, we speculated that the food intake with absences of $LT > NT$ might be singular values caused by mild torpor. Another possible explanation might be that they got sick during the second and third feeding test. Multiple ambient cold exposures increased the risk of sickness, which might decrease their appetite. We apologized that in our initial results, we ignored the effect of possible torpor or sickness on the temperature-induced food intake.

Therefore, in the revised version, we thoroughly checked the food intake data across three times, and discarded data from torpor- or sick- like mice, and performed extra experiments to replenish the corresponding data, specifically, including 2 mice for the ARC, 1 mouse for the BNST in the Figure 3d-f, 2 mice for the control group and 2 mice for the taCasp3 group in the Figure 4c, 2 mice in the Figure 4h, 2 mice in

the Figure 4k. In the revised figures, we added the supplementary data into the figures, replacing the original singular data. Statistics results showed that the LT > NT pattern of food intake was significant after the exclusion of singular data and addition of new data. Furthermore, we have added relevant descriptions about the occurrence of torpor and sickness during cold exposure into the Methods. Mice with significant decreases in food intake, locomotion or rectal temperature after multiple times of cold exposure might turn into mild torpor or sickness and their data should be discarded. We hope that these revisions could improve the data robustness and address your concerns.

Question 2: *Another consideration related to the link between temperature and the roles of these neurons in feeding is their responsiveness to metabolic state. Lines 52-54 state that “In the instances of extremely hot environments, the lower thermal demand and higher dissipation load preferentially suppress food intake, overriding the dietary state.” The effects of fasting or refeeding on cFOS are only measured at room temperature. The effects of ambient temperature on calcium activity are only measured at one ambient temperature. Wouldn't the quoted statement predict that temperature manipulations would alter the effects of fasting on cFOS? This might be helpful if the effects of caspase ablation are unaltered by temperature.*

Answer: We very agree with this insightful comment. In response to your concern, we performed additional experiments to explore the fasting or refeeding-related C-Fos expressions at low or high temperatures. The C-Fos expressions in the apMPOA of mice with different dietary states (*no fasting*, *fasting*, and *refeeding*) during ambient temperature exposures (*NT*, *HT* and *LT*) were compared. As depicted in Figure S3, substantial C-Fos expressions in the apMPOA could be found under the following conditions, including *no fasting*, *fasting* and *refeeding* at low temperature, *no fasting*, *fasting* and *refeeding* at high temperature, *fasting* but not *no fasting* or *refeeding* at neutral temperature. Statistical quantitative analysis showed that interaction effect of temperature and dietary state: $F(4,24) = 5.25$, $**p = 0.003$; main effect of temperature: $F(2,11) = 219.6$, $***p < 0.001$; main effect of dietary state: $F(2,12) = 67.8$, $***p < 0.001$. Pairwise comparisons showed that more C-Fos expressions in the *fasting* condition than *no fasting* and *refeeding* during all the thermal exposures, especially during the neutral thermal exposure. In response to the reviewer's concern on the thermal exposures on the fasting-induced C-Fos, post-hoc comparisons on C-Fos expressions of the fasted mice showed that more C-Fos expression during low temperature than neutral and high temperatures (all $***p < 0.001$). But no differences in C-Fos expression of fasted mice at neutral and high temperatures were found ($p = 0.068$). From the perspective of C-Fos expression, we could not conclude that high temperature inhibit fasting-induced C-Fos with less expressions during high temperature as expected.

We think that possible explanations might be that the apMPOA is enriched with massive thermosensitive neurons, so that the substantial C-Fos expressions in apMPOA might be caused by low or high temperatures, rather than just fasting. Similar results can be also found in the supplementary results of the study by Zheng-

Dong Zhao [5]. In this case, it appears that the effect of fasting and refeeding on C-Fos is masked by the effect of ambient exposure, and it is difficult to distinguish the effect of fasting/refeeding or thermal exposure on C-Fos expression. For example, the feeding behavior of the mice is inhibited in high temperature exposure, and hunger-related C-Fos in the apMPOA might be expected to be reduced due to high temperature exposure. However, the high temperature itself can cause substantial thermosensitive neuronal activations in the apMPOA, so that we can see that under high temperature conditions there are still a lot of C-Fos expressions in the apMPOA in the fasted mice. It is difficult for us to distinguish whether hunger-related C-Fos is reduced or not. Therefore, in our original manuscript, we showed the effect of fasting and refeeding on C-Fos at neutral temperature, but not high or low temperature. In the conditions of neutral temperature, substantial C-Fos expressions might arise mainly from dietary state changes, rather than the effect of temperature.

In the revised manuscript, we have added the expression of C-Fos associated with fasting or refeeding under NT, HT and LT thermal conditions (Figure S3) and added corresponding descriptions in the results and legends. We hope that these experiments and explanations could address your concern.

Minor points:

Question 1: *It is unclear what is meant by: “A similar feeding pattern induced by pharmacological apelin application with optogenetic/chemogenetic manipulations on apMPOA→PVH pathway indicated that the *Aplnr* might be a potential marker for the PVH-projecting apMPOA neurons.” Please rephrase.*

Answer: We apologize for the lack of clarity in our original description. In our original intention, *Aplnr* was a candidate gene for PVH-projecting apMPOA neurons by RNA-sequencing. During further behavioral test, the modulatory effect of pharmacological apelin application on feeding behavior coincided with effect of optogenetic/chemogenetic/caspase ablating manipulations on the apMPOA→PVH pathway (Figure 3d-f, Figure 4c, 4h). The PVH-projecting apMPOA neurons were primarily overlapped with apelin receptor evidenced by further RNAscope experiment. These findings indicated that *Aplnr* might be a potential marker for the PVH-projecting apMPOA neurons.

In the revised manuscript, we have rephrased the sentence as “The modulatory effect of pharmacological apelin application on feeding behavior coincided with optogenetic/chemogenetic/caspase ablating manipulations on apMPOA→PVH pathway (Figure 3d-f, Figure 4 c-h) further indicating that the *Aplnr* might be a potential marker for the PVH-projecting apMPOA neurons”.

Question 2: *All Figures should have legends in the panels, not just in the text.*

Answer: Thank you for pointing out this. During the submission, we have added the corresponding legends when uploading figure files. But there were no legends below the figures in the pdf file exported from the submission system. We have sent this issue to the editorial department to deal with it for better readability.

Question 3: *More methodological clarity is required, such as experimental details, injection volumes, mouse strains, sexes, body weights. For example, body mass can influence all of the variables measured, and so it should be clear that this and other characteristics of the mice are consistent across groups and across experiments.*

Answer: Thank you for pointing out this. In response to your comment, we enriched methodological details that you raised, including experimental details, injection volumes, mouse strains, sexes, body weights. Furthermore, we checked the descriptions of the methods section in detail and enriched it where necessary to improve the methodological clarity.

Response to Reviewer 2:

Question 1: *The selection of the galanin and apelin receptors based on the expression data results seems somewhat arbitrary. Were other neuropeptides tested for an effect on food intake?*

Answer: We thank the reviewer for pointing this out. We apologize for the lack of clarity in the galanin and apelin receptors. We performed gene annotation analysis via KEGG pathway category on the generated 455 DEGs, that were divided into categories including metabolism, environmental information processing, organismal systems, cellular processes. We then selected the top ten genes in each category for further pharmacologic and morphological verifications (Figure 7d-g). Among these selected genes, many were previously not reported to be involved into energy metabolism, thermosensation and regulation that were related to this study. Therefore, we selected the top ten DEGs in each category as potential candidate genes depending on their functions in metabolism-related cellular processing and intercellular transmission activity according to previous studies. For example, two highly expressed genes encoding cellular metabolic-related enzymes among the top ten genes in the environmental information process and metabolism category, the genes *Eno1* and *Aplnr* were upregulated in PVH group (Figure 7d-e). The gene *Aplnr* encodes neuropeptide apelin receptor, which plays important roles in angiogenesis, energy metabolism, and fluid homeostasis [6,7]. The gene *Eno1*, encoding a glycolysis pathway enzyme enolase 1 with multiple functions in cellular energy metabolism, is involved in body weight gain and cumulative food intake [8]. In contrast, the genes *Crhr2* and *Htr1f* were upregulated in the ARC group (Figure 7e). The gene *Crhr2*, encodes corticotropin releasing hormone receptor 2, which contributes to stress-induced anorexia and abnormal body weight [9]. The gene *Htr1f* was selected as candidate gene since serotonin neurons in multiple nuclei are involved in feeding modulation [10,11]. In the organismal systems category, the genes *Drd3* and *Irak4* were upregulated in the PVH group and were selected as candidate genes (Figure 7f). The *Drd3*, especially expressed in striatum neurons, is related to appetite and reward [12,13].

Considering that the POA contains a series of interrelated cell types involved in thermal and energy homeostasis, we also analyzed the expression of genes related to feeding behavior that may not fall into the top ten of data-driven gene annotation analysis described above (Fig. 7h). We compared a range of neuropeptide receptors

associated with feeding [14], including subtypes of Htr, POMC, AgRP, Ghnr, subtypes of MCR, MSH, galanin and NPY receptors (Figure 7k). We found that only the genes *Htr1f*, *NPY1r*, *GalR1* and *NPY5r* showed between-group differences (Figure 7h). In view of these considerations, we selected the DEGs *Htr1f*, *Irak4*, *Aplnr*, *Itgb5*, *Eno1*, *Crhr2*, *Drd3*, *GalR1*, *Npy1r* and *Npy5r* as candidate marker genes for further behavioral validation. The effect of all these candidate genes on feeding behavior was then tested by microinjecting inhibitors or agonists of all these candidate genes into apMPOA (Figure 7i-k). And we found that the genes *Aplnr* and *GalR1* had regulatory effect on food intake. And further morphological evidence (immunofluorescence experiment in the former manuscript and RNAscope experiments in the revised manuscript) verified that ARC-projecting and PVH-projecting apMPOA neurons were molecularly distinct and marked by *GalR1* and *Aplnr*, respectively.

In the revised manuscript, we have enriched the descriptions about selections of candidate genes. We hope that these explanations and revisions could address your concerns.

Question 2: *It would be helpful to know which marker genes differentiate the two apMPOA glutamatergic cell populations from other apMPOA glutamatergic cell populations. It probably would be helpful if the authors could try to map the transcriptional signatures of their two populations onto the Moffit et al single-cell atlas of the preoptic nucleus of the hypothalamus (<https://www.ncbi.nlm.nih.gov/pmc/articles/PMC6482113/>) and show which of the Moffit cell populations their two populations correspond to. This step might also allow them to identify where within the apMPOA their cells are located (Moffit et al provide a spatial atlas along with the single-cell RNA-seq atlas).*

Answer: Thank you for your insightful and constructive comment that could provide strong molecular evidence for our RNA-seq results.

Regarding to the issue about the marker genes differentiate the two apMPOA glutamatergic cell populations from other apMPOA glutamatergic cell populations, we would like to clarify the samples involved into RNA-seq. We retrogradely traced the PVH-projecting and ARC-projecting apMPOA neurons, and then collected them using glass pipette. Therefore, we can only collect the fluorescence labeled glutamatergic neurons, but not other apMPOA glutamatergic cell populations. The other apMPOA glutamatergic cell populations were not involved into RNA sequencing, and we could not determine the marker genes differentiate the two apMPOA glutamatergic cell populations from other apMPOA glutamatergic cell populations. We hope to further clarify this issue in future studies.

Regarding to mapping our transcriptional signatures onto the Moffit et al single-cell atlas of the POA, we didn't obtain their raw RNA-seq and MERFISH data through the published materials. Therefore, we can only make a preliminary analysis based on the data in the figures and tables in their supplementary materials. In the study of Moffitt et al. [15], they thought that neuromodulator receptors were expressed more widely and at lower levels than neuromodulators and transcription factors, with limited use as potential markers for neural function. Therefore, they focused on the molecular and

spatial profiling of neuropeptides and molecules involved in neuromodulator production and transport, as well as for transcript factors, but not for neuromodulator (neuropeptide and hormone) receptors. So we did not get the spatial profiles of our marker genes *Aplnr* and *GalR1* in their study. Despite that, we got the gene expressions of *GalR1* in the both of neuronal excitatory and inhibitory clusters identified in the MERFISH (Table S8 in their supplementary table). The top ten excitatory neuron clusters for *GalR1* gene expression identified by MERFISH were: E21, E19, E6, E29, E9, E13, E20, E12, E7, E1 (Table S8 in their supplementary tables). According to the spatial distributions of the neuronal clusters in the brain atlas (see the Figure S17 and Figure S18 in their supplementary materials), the neuronal clusters E1, E2, E3, E6, E11, E20 were primarily enriched in the apMPOA (including AVPe, VMPO, MnPO and periventricular hypothalamic nucleus) that we concern. The overlapped clusters between top ten excitatory clusters of *GalR1* and spatially enriched clusters in apMPOA were E1, E6, E20. In their supplementary Table S9, notable genes observed in the clusters E1, E6, E20 were *Glut* (in the cluster E1), *Nos1/Trp73* (in the cluster E6), *Crh* (in the cluster E20) using the MERFISH. The marker gene *GalR1* in our study probably corresponded to these genes identified by Moffit et al. However, due to lack of spatial distribution of gene *GalR1* in the study by Moffit et al, we can only make some speculations based on the data they provided, and these speculations should be cautious before identification. Regarding to our maker gene *Aplnr*, we did not find relevant data in the study by Moffit et al. They stated that neuromodulator receptors were not involved into scRNA-seq or MERFISH due to wide but low expressions and limited use as potential markers for neural functions, therefore, the *Aplnr* might not be included in the cluster analysis of their study. In the revised version, we have performed RNAscope experiments raised by another reviewer to further identify the RNA expressions of the genes *GalR1* and *Aplnr* and spatial colocalizations with fasting-induced C-Fos. These experiments would further support the spatial distributions and neuronal functions of the genes *GalR1* and *Aplnr*. These results were added in the revised Figure 7.

We hope that these explanations and experiments could satisfy your concerns.

Question 3: *In general, a bit more information on these cell populations' transcriptome would be helpful. How many unique genes where identify for each of the two cell populations? How many gene sets were tested for enrichment? Where other databases then GO and KEGG tested? How were there adjusted for multiple testing?*

Answer: Thank you for your helpful suggestion, we apologize for lack of sufficient information about RNA-seq. A total of 14917 transcripts were co-expressed in the PVH and ARC-projecting apMPOA neurons. And 765 and 972 transcripts were uniquely expressed in the PVH-projecting and ARC-projecting apMPOA neurons, respectively. In the revised version, we have added the Venn diagram depicting the number of genes co-expressed and uniquely expressed between the PVH and ARC-projecting apMPOA neurons. We only used the differentially expressed genes between the PVH and ARC-projecting apMPOA neurons as KEGG pathway category

analysis. In this study, we only did KEGG analysis, and did not use other annotation databases.

According to the KEGG_pathway annotation classification, we used the Phyper function in R package to perform the enrichment analysis (<https://stat.ethz.ch/R-manual/R-devel/library/stats/html/Hypergeometric.html>), and calculated the P value. And the Q value was obtained by the resulting P -values that were adjusted for controlling the false discovery rate using the Benjamini and Hochberg's approach (<https://bioconductor.org/packages/release/bioc/html/qvalue.html>). The function of Q value ≤ 0.05 is regarded as a significant enrichment.

Question 4: *The same applies for the RNA sequencing part of the Methods. What was the sequencing depth? Is the “Dr. Tom system” part of the BGI software suite? Which version of KEGG and GO were used? What was the gene background used in the gene set enrichment analysis?*

Answer: Thank you for pointing out the insufficient information about RNA-seq. Dr. Tom system is an interactive report system developed by BGI. In this project, a total of 10 samples were measured, the libraries were multiplexed and sequenced on an DNBSEQ platform, and 100-bp paired-end reads were generated, and each sample produced an average of 6.83G data. For transcript abundance quantification analysis, HISAT2 (v2.0.4) was used to map the reads to the mouse genome (mm10, build name GRCm38). The version of KEGG pathway analysis was 93.0 obtained from KEGG database (93.0/genes/ko/ko_pathway.list). And we did not perform gene set enrichment analysis or gene ontology (GO) analysis in our original manuscript.

In the revised version, we have updated the above information in the Methods section.

Question 5: *The names of all differentially expressed genes should be made available along with their test statistics and normalized expression values.*

All gene set enrichment results should be made available as source data.

All expression data should be made available.

All source code used to process and analyze the data should be made available to allow others to reproduce their results and to learn from their setup.

Answer: Thank you for your helpful suggestion. In response to it, in the revised version, we have added the expression data of all the transcripts in the supplementary Table S1, and the names, expression data and statistics of all the DEGs in the supplementary Table S2. The original RNA-seq data of PVH and ARC-projecting apMPOA neurons have been deposited in the database of the NCBI Sequence Read Archive (<https://www.ncbi.nlm.nih.gov/sra/>) under the accession number PRJNA820979.

Response to Reviewer 3:

Major comments

Question 1: *Can you exclude that axonal stimulation leads to backpropagation of activation and spread to other projection areas (thus are there multiple innervation targets from a single neuron?)*

Answer: Thank you for your insightful comment. We very agree that terminal activation would lead to action potential back propagation to other areas [16]. Regarding this issue, we carried out additional experiments on the following three aspects.

First, we silenced the upstream apMPOA by injecting muscimol when optogenetic activating fiber terminals in one downstream target to prevent the back propagation to other downstream targets. However, we found that silencing the apMPOA by injecting muscimol lead to paralysis and hypothermia in all the mice. Although this effect was reversible, it took the 5-6 hours for mice to recover. One possible reason could be that MPO has multiple essential physiological functions related to life support, such as body temperature regulation, locomotion, etc. This made it difficult to observe the feeding behavior through silencing MPO. Therefore, we abandoned this approach to exclude the back propagation and tried the other two aspects.

In response to exclusion of back propagation, we think that it would be helpful to investigate the existence of a single apMPOA neuron innervating multiple targets as you raised in the ***Question 10 in Other Comments***. Therefore, we performed extra experiment by injecting retrograde AAVretro-DIO-GFP and AAVretro-DIO-mCherry into the downstream targets PVH and ARC of vGluT2-cre and GAD2-cre mice that we concentrated in this study. This can be also seen in the response to ***Question 10 in Other Comments***. We found that only a small fraction of colocalized PVH and ARC-projecting neurons in apMPOA were found in the GAD2-cre mice, but not in the vGluT2-cre mice that we concentrated in this study. This finding suggested that two separated glutamatergic apMPOA cell populations innervated the PVH and ARC, and our original terminal activation on glutamatergic apMPOA neurons in the downstream regions was unlikely to generate back propagation. We have added this result in the supplementary Figure S10. This retrograde tracing experiment was also applied on other downstream regions, including PAG, DMH, LH, and BNST. And we did not find substantial overlap retrograde labeled neurons in apMPOA from pairwise downstream regions.

Thirdly, we further verified our original behavioral results of feeding behavior under terminal activation through postsynaptic activation by using high titer of AAV1 virus. We performed optogenetic stimulation on apMPOA-recipient postsynaptic neurons in the ARC, PVH by injecting high titer of AAV1-cre in the apMPOA and AAV-DIO-ChR2 in the PVH and ARC. We found that the effect of postsynaptic activation on feeding behavior were similar with that of terminal activation. The optogenetic activation on apMPOA-recipient ARC neurons increased food intake. In contrast, postsynaptic activation on the PVH reduced food intake. We have added this result in the supplementary Figure S5.

Taken together, we thought that two separated cell populations innervated the PVH

and ARC, and terminal activation were unlikely to induce action potential back propagation. Moreover, we further identified our original results by using high titer of AAV1 virus. In the revised manuscript, we have added the results of optogenetic activation on postsynaptic neurons and colocalizations of retrograde PVH and ARC-projecting apMPOA neurons into the supplementary Figure S5 and Figure S10. We hope that these support findings could address your concerns.

Question 2: *The data in figure 4 require more controls. The authors claim AAVcreGFP jumps anterogradely but there are several alternative explanations. It could also be that this virus is taken up by terminals and is backtransported to the nucleus of cells in the MPO target areas (thus neurons projecting to MPO may have been targeted). There may be very limited leaky expression in absence of cre.*

Answer: Thank you for your insightful comment with which we very much agree. Indeed, high titer of AAV1, as an anterograde transsynaptic tool for tagging postsynaptically targeted neurons, is probably taken up by terminals and is backtransported to the upstream nucleus. This is well-demonstrated in the study by Zingg et al. [16]. Therefore, we had performed several control experiments prior to the higher-titer AAV1 experiment in Figure 4. We injected anterograde AAV-DIO-GFP into the PVH or ARC of vGluT2-cre mice to investigate whether there are terminals in the apMPOA innervated by PVH or ARC. The result showed that no projections from PVH to apMPOA, and or very sparse projections from ARC to apMPOA. These results are also consistent with publicly accessible data from the Allen brain atlas (such as Experiment 146554676 and 302221478, <http://connectivity.brain-map.org/>). In the revised manuscript, we have added these results in the supplementary Figure S6a-e.

Additionally, we injected retrograde AAVretro-DIO-GFP in the apMPOA of vGluT2-cre mice to investigate the presence of retrogradely labeled neurons in PVH or ARC. The results showed that there were no retrograde labelled neurons in the PVH or ARC from the apMPOA. The results further demonstrated there were no substantial projection from PVH or ARC to apMPOA. In the revised manuscript, we have added these results in the supplementary Figure S6f-i.

In the revised version, we have added these control results and related descriptions in the methodological supplementary materials. We hope that these explanations and experiments will address your concerns.

Question 3: *Also for the Rabies experiments controls are needed without TVA injected in Glut-cre mice. Also all virus should be injected in wt mice to exclude leaky expression*

Answer: Thank you for your helpful suggestion. We have now carried out extra experiments to address this point. For the control experiment, we injected the helper virus AAVretro-DIO-oRVG without AAVretro-DIO-TVA-EGFP unilaterally into the ARC or PVH of vGluT2-cre mice. After expression for three weeks, the rabies virus RV-ENVA- Δ G-dsRed was injected into the apMPOA. Then, the mice were returned to their home cages. After one week of viral replication, the mice were sacrificed and

their brains were prepared for further imaging. The results showed no dsRed-labeled neurons in the apMPOA or other areas in the brain, indicating the dependence of the RV infection on the TVA expression.

Additionally, we performed control experiments in wild-type mice. We injected the mixed helper viruses with AAVretro-DIO-oRVG and AAVretro-DIO-TVA-EGFP unilaterally into the ARC or PVH of wild-type mice. After three weeks, the rabies virus RV-ENVA- Δ G-dsRed (100 nL) was injected into the apMPOA. The results showed no TVA-GFP or RV-dsRed expression in the injection sites or apMPOA, suggesting no leaky expression of the helper viruses or RV.

In the revised manuscript, we have added these control experiments in the method section and supplementary Figure S12.

Question 4: *I recommend RNASCOPE experiments to verify VGlut, apelin receptor, galanin receptor in cells that get fos activated by fasting combined with projection specific markers to verify the identity of the projection specific glutamatergic neurons*

Answer: Again, we sincerely thank you for your suggestions to make this study more solid. To address this concern, we performed extra RNAscope experiments instead of original immunofluorescence experiments. As before, we labeled the projection specific glutamatergic apMPOA neurons by injecting Cre-dependent AAVretro vectors expressing mCherry into the PVH or ARC (not in the same mouse) of vGluT2-cre mice. Four mice were used in each group. After expression for 3 weeks, the mice were overnight fasted and sacrificed for further RNAscope experiments. The FISH was performed with probes targeting *aplnr* (GenBank: NM_011784.3, RNAscope®Probe-Mm-Aplnr), *GalR1* (GenBank: NM_008082.2, RNAscope®Probe-Mm-Galr1-C2) and *Fos* (GenBank: NM_010234.2, RNAscope®Probe-Mm-Fos-C3) mRNA by using the RNAscope kit (Advanced Cell Diagnostics) as indicated in user manual. The results of colocalizations replicated our original findings of immunofluorescence experiments. Specifically, The ARC-projecting neurons in the apMPOA exhibited substantial colocalization with *GalR1*. Approximately 53% of ARC-projecting neurons expressed galanin receptor 1 (revised Figure 7m-n), but minor colocalization with apelin receptor (approximately 6.9%, revised Figure S13a-b). Approximately 40.5% of the PVH-projecting apMPOA neurons were overlapped with apelin receptor (revised Figure 7o-p), but fewer with galanin receptor 1 (approximately 20%, revised Figure S13c-d). These findings indicated that the two subsets of ARC-projecting and PVH-projecting apMPOA neurons were molecularly distinct. The fasting-induced C-Fos showed higher colocalization with ARC-projecting neurons and galanin receptor 1 (approximately 30% of C-Fos positive neurons, revised Figure 7m-n, Figure S13a-b), but less overlap with PVH-projecting neurons and apelin receptor (approximately 12.5%, revised Figure 7o-p, Figure S13c-d). More ARC-projecting neurons (approximately 42.5% ARC-projecting neurons, revised Figure 7m-n and Figure S13a-b,) than PVH-projecting labeled neurons (approximately 12.5% PVH-projecting neurons, revised Figure 7o-p and Figure S13c-d,) were colocalized with the fasting-induced C-Fos. This suggested that hunger-related neurons in apMPOA are highly overlapped with

galanin receptor 1 and ARC-projecting neurons. Detailed information about quantifications can be seen in the revised Figure 7m-p and Figure S13a-d. In the revised manuscript, we have enriched the RNAscope related descriptions in the Result, Method and Discussion sections.

Question 5: *For all experiments details on success of targeting nuclei with viruses are missing. How many mice were excluded from the experiments due to missed injections? Were there unilateral hit mice when the purpose was bilateral hit? What was the total amount of mice used for these experiments?*

Answer: We apologize for the lack of details regarding the number of animals. There were failures of virus injection, cannula and fiber implantation, sickness or death after surgery and incomplete data during test in the experiments. In the revised manuscript, we have enriched the detailed information on relevant failures and the number of mice in the figure legends.

Other comments

Question 1: *The grammar of the manuscript needs to be improved by a native English speaking scientist. In some cases poor language results in wrong statements such as in line 408 “...which induce proopiomelanocortin (POMC) neurons....” Which does not make sense. Which are activated by POMC neurons is probably meant.*

Answer: Thank you for your helpful advice. We have carefully corrected the grammatical errors and poorly organized sentences with the help of a native English speaker. We hope that all the corrections could reduce misleading points of the manuscript. All the corrections were highlighted in yellow in the revised manuscript.

Question 2: *How was proximity between cFos and ChR axons determined*

Answer: Thank you for pointing out this issue, which is related to Figure 3b and Figure S3b. In both figures, we intent to show the downstream regions of apMPOA by anterograde labeling using ChR2 axons. In addition, we deprived the mice of food overnight to induce C-Fos activations in multiple brain regions, including those downstream regions that apMPOA projected. Our initial intention was to show these apMPOA-innervated downstream regions were related to fasting. Therefore, we did not calculate the proximity between C-Fos and axons here. This method was also used in the study by Alhadeff et al. [17]. We apologize that the original sentence “We quantified neurons directly under apMPOA^{vGluT2+} axons that expressed hunger-induced C-Fos” was not well organized and misleading to you. In the revised manuscript, we rephrased the sentence as “We quantified neurons that expressed hunger-induced C-Fos within the downstream regions that apMPOA^{vGluT2+} neurons projected”. We hope that the revised description could minimize the misinformation.

Question 3: *To which cells in arc do Glu neurons project to, likely AgRP as FI was increased?*

Answer: Thank you for your insightful comment. In our study, we focused on the glutamatergic neurons in apMPOA and their projections to ARC, whereas the cell

types of postsynaptic neurons in ARC were not intentionally explored. In this study, we had bred vGluT2-cre and GAD2-cre mice with Ai9 reporter mice, and observed the distributions of excitatory and inhibitory neurons in the ARC. We found substantial GAD2⁺ neurons in ARC, but sparse vGluT2⁺ neurons as depicted in following figure. This can be also found in the Allen brain atlas (Experiment 73818754, <http://mouse.brain-map.org/experiment/show/73818754>; Experiment 79591669, <http://mouse.brain-map.org/experiment/show/79591669>). Moreover, the specific cell types have been well illustrated in other previous studies. In two recent studies, Deem et al. [18] and Yang et al. [1] found that AgRP neurons activations occurs rapidly upon acute cold exposure and promote food intake, while silencing of AgRP neurons selectively blocked the cold exposure to increase food intake. Considering those previous findings, we did not explore the postsynaptic neurons in ARC. In the revised manuscript, we have added these previous findings into Discussion to further clarify the neural circuit for temperature-regulated feeding.

Figure The vGluT2⁺ and GAD2⁺ neurons distribution in the ARC

Question 4: For the SmartSeq experiments what is a biological replicate? Is it 5 mice?

Why was FACS sorting not done in order to perform single cell level analyses?

Answer: In the SmartSeq experiment, we collected RNA profiles from 5 biological replicates in each group. This meant that 5 mice were used for each group.

For concern about cell collections, in order to accurately collect retrograde labeled neurons in apMPOA rather than neighboring nuclei, we collected fluorescence labeled neurons in acute sections with a glass pipette after retrograde labeling, rather than with FACS. In the retrograde labeling experiment, we injected a Cre-dependent AAVretro vector expressing EGFP into the PVH or ARC (not in the same mouse) of vGluT2-cre mice (Fig. 7a). The results showed that not only fluorescence labeled neurons were present in apMPOA, but also in some very neighboring nuclei, such as BNST, MPN. If FACS was used, it will be difficult to retain only the labeled neurons in the apMPOA, not the other neighboring nuclei. While using the electrophysiological platform, the fluorescently labeled neurons could be visualized under a microscope, and be collected accurately in apMPOA, rather than other nuclei. Therefore, for precise collection, we referred to the study by Chen et al. [19], and collected the cells under a microscope with a glass pipette. Although this approach is

less efficient and technically challenging, the labeled cells are in a better state since the oxygen-saturated ACSF was continuously perfused. We hope that these explanations will address the reviewer's concerns. In the revised manuscript, we have added a brief explanation for the approach for cell collection in the Method section.

Question 5: *The discussion should be written more condense and focus on the most important findings. There is also some overinterpretation such as: the LPB-MPOA-ARC pathway was anatomically mapped, but the authors did not demonstrate that LPB provide temperature information to the neurons engaged in feeding responses. The only show connectivity.*

Line 435: "These findings revealed an unrecognized neural circuits of temperature-regulated feeding behavior orchestrated by LPB→apMPOA→PVH/ARC pathways" is such overinterpretation.

Answer: Thank you for your helpful comment with which we very agree. The main finding of this study was apMPOA→PVH/ARC pathways for modulating feeding behavior in coping with ambient temperatures. In response to your comment, we added more discussions temperature-regulated feeding behavior related to apMPOA→PVH/ARC pathways. Specifically, we added three recent studies on neuronal responses of AgRP neurons in the ARC to thermal stimulus as following: "The apMPOA→PVH and apMPOA→ARC, presenting warm and cold thermosensitive properties, might play mutually antagonistic roles in the regulation of feeding. In two recent studies, Deem et al. [18] and Yang et al. [1] found that AgRP neurons activation occurred rapidly upon acute cold exposure and promote food intake, while silencing of AgRP neurons selectively blocked the cold exposure to increase food intake. While Zimmer et al. [20] found that thermal stimulus in neonates modulated the functional properties of AgRP neurons in the ARC, with warm temperature blunting the activations of AgRP neurons."

Also, we further simplified the discussions about the upstream pathway LPB→apMPOA and emphasized that the findings of this paper mainly focus on anatomical connections, while further studies are still needed for functional manipulations. Some overinterpreted and repeated sentences were deleted, such as the original sentences "The apMPOA comprises multiple cell populations, and innervates downstream targets that mediate complex innate behaviors, and homeostatic regulatory processes", "Supporting this, activation on apMPOA neurons induced both temperature and feeding behavior alterations", "These findings revealed an unrecognized neural circuits of temperature-regulated feeding behavior orchestrated by LPB→apMPOA→PVH/ARC pathways"... We hope that these revisions could address your concern.

Question 6: *Experimental details are missing such as injected volume of Cre-dependent hChR2 virus (AAV2/8-EF1α-DIO-hChR2-EYFP)*

Since 40C is warm but not hot, I propose to replace hot plate with warm plate (as hot plate is also used to induce pain).

Answer: Thank you for pointing out this. We have checked the description of the

methods section in detail and enriched it where necessary.

We agree that the phrase “warm plate” is more appropriate. In the revised version, we have corrected the phrase in the manuscript.

Question 7: *Concentrations of CTB-555 and CTB-488 are not mentioned*

Answer: We apologize for lack of information of CTB-555 and CTB-488. The CTB-555 (Invitrogen, C34776) and CTB-488 (Invitrogen, C-34775) were dissolved in PBS as 1.0mg/mL that was recommended in the product manuals. We have enriched this information in the Method.

Question 8: *For the optogenetic stimulation in fig 2 what were the parameters (light intensity, frequency etc)*

Answer: Again, we apologize for the lack of experiment details in the original manuscript. The parameters of 473 nm laser for optogenetic activation were set as: 10 ms light pulse at 20 Hz in 1s ON and 3 s OFF duty cycle, 5 mW at fiber tip. The duration of optogenetic stimulation: 2 h for the feeding test, 60 min for the rectal temperature recording, and 30 min for the infrared thermography. In the revised version, we have enriched these descriptions in the Method.

Question 9: *In fig 3 were the same mice used at different ambient temperatures? What were the conditions for optostim?*

Answer: Indeed, in the optogenetic regulation of feeding behavior shown in Figure 3, the same mice were repeatedly measured at different ambient temperatures with 3-5 days interval. In the revised manuscript, we have further clarified this point.

The parameters of optogenetic stimulation were set as: 10 ms light pulse at 20 Hz in 1s ON and 3 s OFF duty cycle with 5 mW. The duration of optogenetic stimulation: 2 h for the feeding test.

Question 10: *Fig 6c suggest that a single neuron innervates PVN and Arc, how does this impact on the results of optostim of target regions?*

Answer: Thank you for your insightful comment.

In the Figure 6c, we found that retrograde-labeled neurons from PVH and ARC were sparsely colocalized with each other, but still a small fraction of colocalization (approximately 14%). As you mentioned, there are indeed cases where there are single neurons innervated by both PVH and ARC. Since the CTB we used here was not a cell type-specific retrograde tracer, the retrograded labeled neurons in apMPOA might not all be the glutamatergic neurons of interest in our study, but might be other types of neurons. According to previous studies, the medial preoptic are is mainly composed of glutamatergic and GABAergic neurons [15]. As the **Answer to Question 1**, we performed extra experiments by injecting Cre-dependent AAVretro-DIO-GFP and AAVretro-DIO-mCherry into the ARC and PVH in the vGluT2-cre and GAD2-cre mice. We found that only a small fraction of colocalized PVH and ARC-projecting neurons in apMPOA were found in the GAD2-cre mice, but not in the vGluT2-cre mice that we concentrated in this study (Figure S10). This finding suggested that two

separated glutamatergic apMPOA cell populations innervated the PVH and ARC, and our original terminal activation on glutamatergic apMPOA neurons in the downstream regions was unlikely to generate back propagation. We have added this result in the supplementary Figure S10.

Question 11: *The title of Fig. S2 is misleading: C-Fos expressions in multiple thermosensitive preoptic and hypothalamic nuclei in a hunger state; the authors did not determine in this study whether they were thermosensitive.*

Answer: Thank you for pointing out this misleading title. In the Figure S2, we indeed did not determine whether these nuclei were thermosensitive. In the revised version, we have deleted this word.

1. Yang S, Tan YL, Wu X, Wang J, Sun J, et al. (2021) An mPOA-ARCAgRP pathway modulates cold-evoked eating behavior. *Cell Rep* 36: 109502.
2. Hrvatin S, Sun S, Wilcox OF, Yao H, Lavin-Peter AJ, et al. (2020) Neurons that regulate mouse torpor. *Nature* 583: 115-121.
3. Takahashi TM, Sunagawa GA, Soya S, Abe M, Sakurai K, et al. (2020) A discrete neuronal circuit induces a hibernation-like state in rodents. *Nature* 583: 109-114.
4. Zhang Z, Reis F, He Y, Park JW, DiVittorio JR, et al. (2020) Estrogen-sensitive medial preoptic area neurons coordinate torpor in mice. *Nat Commun* 11: 6378.
5. Zheng-Dong Zhao WZY, Cuicui Gao, Xin Fu, Wen Zhang, Qian Zhou, Wanpeng Chen, Xinyan Ni, Jun-Kai Lin, Juan Yang, Xiao-Hong Xu, Wei L. Shen (2017) A hypothalamic circuit that controls body temperature. *Proc Natl Acad Sci U S A* 114: E1755.
6. Chapman NA, Dupre DJ, Rainey JK (2014) The apelin receptor: physiology, pathology, cell signalling, and ligand modulation of a peptide-activated class A GPCR. *Biochem Cell Biol* 92: 431-440.
7. De Mota N, Reaux-Le Goazigo A, El Messari S, Chartrel N, Roesch D, et al. (2004) Apelin,

- a potent diuretic neuropeptide counteracting vasopressin actions through inhibition of vasopressin neuron activity and vasopressin release. Proc Natl Acad Sci U S A 101: 10464-10469.*
8. *Cho H, Lee JH, Um J, Kim S, Kim Y, et al. (2019) ENOblock inhibits the pathology of diet-induced obesity. Sci Rep 9: 493.*
 9. *Makino S, Asaba K, Nishiyama M, Hashimoto K (1999) Decreased type 2 corticotropin-releasing hormone receptor mRNA expression in the ventromedial hypothalamus during repeated immobilization stress. Neuroendocrinology 70: 160-167.*
 10. *D'Agostino G, Lyons D, Cristiano C, Lettieri M, Olarte-Sanchez C, et al. (2018) Nucleus of the Solitary Tract Serotonin 5-HT_{2C} Receptors Modulate Food Intake. Cell Metab 28: 619-630 e615.*
 11. *Price AE, Anastasio NC, Stutz SJ, Hommel JD, Cunningham KA (2018) Serotonin 5-HT_{2C} Receptor Activation Suppresses Binge Intake and the Reinforcing and Motivational Properties of High-Fat Food. Front Pharmacol 9: 821.*
 12. *Nathan PJ, O'Neill BV, Mogg K, Bradley BP, Beaver J, et al. (2012) The effects of the dopamine D(3) receptor antagonist GSK598809 on attentional bias to palatable food cues in overweight and obese subjects. Int J Neuropsychopharmacol 15: 149-161.*
 13. *van de Giessen E, de Bruin K, la Fleur SE, van den Brink W, Booij J (2012) Triple monoamine inhibitor tesofensine decreases food intake, body weight, and striatal dopamine D₂/D₃ receptor availability in diet-induced obese rats. European Neuropsychopharmacology 22: 290-299.*
 14. *Sohn JW, Elmquist JK, Williams KW (2013) Neuronal circuits that regulate feeding*

behavior and metabolism. Trends Neurosci 36: 504-512.

15. Moffitt JR, Bambah-Mukku D, Eichhorn SW, Vaughn E, Shekhar K, et al. (2018) Molecular, spatial, and functional single-cell profiling of the hypothalamic preoptic region. *Science* 362.
16. Zingg B, Chou XL, Zhang ZG, Mesik L, Liang F, et al. (2017) AAV-Mediated Anterograde Transsynaptic Tagging: Mapping Corticocollicular Input-Defined Neural Pathways for Defense Behaviors. *Neuron* 93: 33-47.
17. Alhadeff AL, Su Z, Hernandez E, Klima ML, Phillips SZ, et al. (2018) A Neural Circuit for the Suppression of Pain by a Competing Need State. *Cell* 173: 140-152 e115.
18. Deem JD, Faber CL, Pedersen C, Phan BA, Larsen SA, et al. (2020) Cold-induced hyperphagia requires AgRP neuron activation in mice. *Elife* 9.
19. Chen Q, Deister CA, Gao X, Guo B, Lynn-Jones T, et al. (2020) Dysfunction of cortical GABAergic neurons leads to sensory hyper-reactivity in a Shank3 mouse model of ASD. *Nat Neurosci* 23: 520-532.
20. Zimmer MR, Fonseca AHO, Iyilikci O, Pra RD, Dietrich MO (2019) Functional Ontogeny of Hypothalamic Agrp Neurons in Neonatal Mouse Behaviors. *Cell* 178: 44-59 e47.

REVIEWERS' COMMENTS

Reviewer #1 (Remarks to the Author):

The authors have adequately addressed my initial concerns, adding data, controls, and statistical details to greatly strengthen the conclusions of the paper.

Reviewer #2 (Remarks to the Author):

The authors addressed all of my previously stated comments and concerns.

Response to Reviewer 1:

Comment: *The authors have adequately addressed my initial concerns, adding data, controls, and statistical details to greatly strengthen the conclusions of the paper.*

Answer: We thank the reviewer for the helpful comments on data robustness, statistical details and control experiments that greatly improved our manuscript. We appreciate the reviewer's kind help for this study.

Response to Reviewer 2:

Comment: *The authors addressed all of my previously stated comments and concerns.*

Answer: We thank the reviewer for the helpful comments on RNA-seq that greatly enriched our RNA-seq results. We appreciate the reviewer's kind help for this study.

Response to Reviewer 3:

Comment: *I thank the authors for taking so much care to address my concerns and performing additional experiments. I have no further comments.*

Answer: We thank the reviewer for the helpful comments on methodological details that greatly improved our data reliability. We appreciate the reviewer's kind help for this study.